# Fighting Fire with Fire: The Dual Role of LLMs in Crafting and Detecting Elusive Disinformation

**Jason Lucas**[1]    **Adaku Uchendu**[1,2]    **Michiharu Yamashita**[1]
**Jooyoung Lee**[1]    **Shaurya Rohatgi**[1]    **Dongwon Lee**[1]

[1] The Pennsylvania State University, University Park, PA, USA
{jsl5710, michiharu, jfl5838, szr207, dongwon}@psu.edu
[2] MIT Lincoln Laboratory, Lexington, MA, USA
adaku.uchendu@ll.mit.edu

## Abstract

Recent ubiquity and disruptive impacts of large language models (LLMs) have raised concerns about their potential to be misused (*i.e.*, generating large-scale harmful and misleading content). To combat this emerging risk of LLMs, we propose a novel "**Fighting Fire with Fire**" (F3) strategy that harnesses modern LLMs' generative and emergent reasoning capabilities to counter human-written and LLM-generated disinformation. First, we leverage GPT-3.5-turbo to synthesize authentic and deceptive LLM-generated content through paraphrase-based and perturbation-based prefix-style prompts, respectively. Second, we apply zero-shot in-context semantic reasoning techniques with cloze-style prompts to discern genuine from deceptive posts & news articles. In our extensive experiments, we observe GPT-3.5-turbo's zero-shot superiority for both in-distribution and out-of-distribution datasets, where GPT-3.5-turbo consistently achieved accuracy at 68-72%, unlike the decline observed in previous customized and fine-tuned disinformation detectors. Our codebase and dataset are available at https://github.com/mickeymst/F3.

## 1 Introduction

While recently published LLMs have demonstrated outstanding performances in diverse tasks such as human dialogue, natural language understanding (NLU), and natural language generation (NLG), they can also be maliciously used to generate highly realistic but hostile content even with protective guardrails, especially *disinformation* (Spitale et al., 2023; Sadasivan et al., 2023; De Angelis et al., 2023; Kojima et al., 2022). Moreover, LLMs can produce persuasive texts that are not easily distinguishable from human-written ones (Uchendu et al., 2021; Chakraborty et al., 2023; Zhou et al., 2023), making humans more susceptible to the intrinsic/extrinsic hallucination proclivities and thus introducing disinformation (Uchendu et al., 2023).

To mitigate this muddle of disinformation by LLMs, in this work, we ask a pivotal question: if LLMs can generate disinformation (via malicious use or hallucination), can they also detect their own, as well as human-authored disinformation? Emerging literature provides limited perspectives on the potential use of the latest commercial state-of-the-art (SOTA) LLMs such as GPT-4 (OpenAI, 2023) and LLaMA 2 (Touvron et al., 2023) to address disinformation. Particularly, topics including: (1) leveraging prompt-engineering to bypass LLMs' protective guard-rails; (2) utilizing the emergent zero-shot capabilities of modern LLMs (with 10B+ parameters) for disinformation generation and detection; (3) manipulating human-written real news to fabricate LLM-generated real and fake narratives to simulate real-world disinformation risks; and (4) assessing and addressing LLMs' inherent hallucinations in the disinformation domain.

To investigate these inquiries, we formulate two research questions (RQs) as follows:

**RQ1: Can LLMs be exploited to efficiently generate disinformation using prompt engineering?**, where we (1) attempt to override GPT-3.5's alignment in fabricating real news, and (2) measure the frequency and remove GPT-3.5 hallucinated misalignments.

**RQ2: How proficient are LLMs in detecting disinformation?**, where we evaluate the capability to detect disinformation between (1) human and AI-authored, (2) self-generated and other LLM-generated, (3) social media posts and news articles, (4) in-distribution and out-of-distribution, and (5) zero-shot LLMs and domain-related detectors.

Pretrained LLMs of our interest are GPT-3.5-Turbo, LLaMA-2-Chat (Rozière et al., 2023),

| Models | Size | Max Token | Source |
|---|---|---|---|
| GPT3.5-Turbo | 175B | 8,192 | OpenAI |
| LLaMA-2-Chat | 70B | 4,096 | Hugging Face |
| LLaMA-2-GPT4 | 70B | 4,096 | Hugging Face |
| Dolly-2 | 12B | 4,096 | Hugging Face |
| Palm-2 | 340B | 8,192 | Hugging Face |

Table 1: Summary of LLMs used.

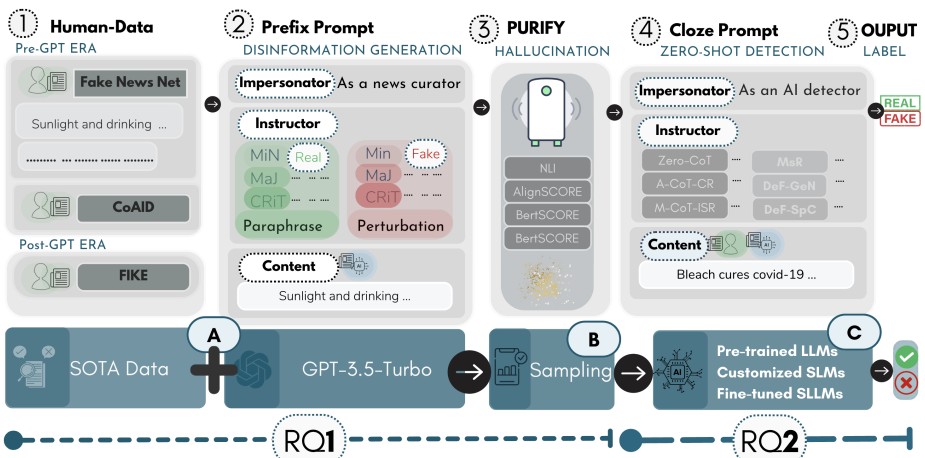

Figure 1: Fighting Fire with Fire (F3) Framework for (A) Disinformation Generation (B) hallucination-purification [detection and removal] (C) In-context Semantic Zero-shot Detection

LLaMA-2-GPT4 (Touvron et al., 2023), Palm-2-text-bison (Anil et al., 2023), and Dolly-2 (Conover et al., 2023) (See Table 1 for more details). To answer these RQs, we propose the **Fighting Fire with Fire (F3)** Framework. As shown in Fig. 1, we first use paraphrase and perturbation methods with prefix-style prompts to create synthetic disinformation from verified human-written real news (steps 1 and 2). We then employ hallucination mitigation and validation strategies to ensure our dataset remains grounded in factual sources. Specifically, the PURIFY (Prompt Unraveling and Removing Interface for Fabricated Hallucinations Yarns) method in step 3 incorporates metrics like AlignScore, Natural Language Inference, Semantic Distance, and BERTScores to ensure data integrity and fidelity. Lastly, steps 4 and 5 implement cutting-edge in-context, zero-shot semantic reasoning such as Auto-Chain of Thoughts for detecting disinformation.

Our contributions include: (1) new prompting methods for synthetic disinformation generation; (2) hallucination synthetic disinformation purification framework; (3) novel prompting in-context semantic zero-shot detection strategies for human/AI disinformation; (4) comprehensive benchmark of SOTA detection models on human/AI disinformation dataset; and (5) dataset for disinformation research. This dual-perspective study cautions the risks of AI disinformation proliferation while also providing promising techniques leveraging LLM capabilities for enhanced detection.

## 2 Related Work

### 2.1 Prompt-based Learning

Latest large language models (LLMs) surpass previous models in many downstream tasks, including zero-shot NLP via prompt engineering (Holtzman et al., 2019; Jason et al., 2022). We survey two core prompting techniques that we use for our task.

**Prefix Prompts** provide instructional context at the beginning of the prompt to guide LLMs' text generation (Kojima et al., 2022). Strategies like in-context learning and prompt tuning boost generative performance on many NLP tasks such as summarization, and translation (Radford et al., 2019; Li and Liang, 2021; Dou et al., 2020; Brown et al., 2020). In addition, paraphrasing (Krishna et al., 2023; Kovatchev, 2022) and perturbation (Chen et al., 2023) approaches are widely used in NLP tasks. We use both paraphrasing and perturbation with prefix prompts to synthetically generate disinformation variations from human-written true news (Karpinska et al., 2022; Fomicheva and Specia, 2019). This leverages LLMs' generation while maintaining its connection to truth.

**Cloze Prompts** contain missing words for LLMs to fill in using context (Hambardzumyan et al., 2021) and are often used to assess LLMs' contextual prediction. This includes question answering to predict the correct missing word that logically completes a given context (Gao et al., 2020). Researchers have applied fixed-prompt tuning to cloze prompts (Lester et al., 2021; Schick and Schütze, 2021). Prior work explored cloze prompt engineering (Hambardzumyan et al., 2021; Gao et al., 2020). We combine cloze prompts with SOTA reasoning techniques such as Chain-of-Thought (CoT) for zero-shot disinformation detection (Tang et al., 2023a), leveraging both approaches.

### 2.2 Disinformation Detection

Earlier disinformation detectors use diverse approaches such as neural, hierarchical, ensemble-

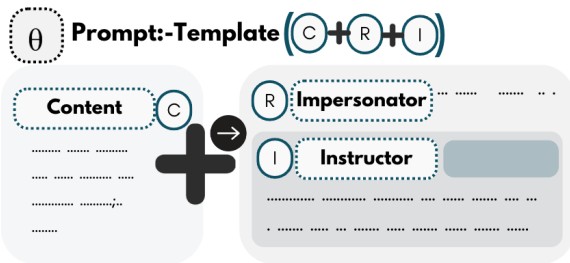

Figure 2: F3 prompt template $(\theta)$ has three parameters: (1) content $(C)$ embeds data. (2) Impersonator $(R)$ establishes context, guides LLMs' generation and detection, and overrides alignment-tuning. (3) Instructor $(I)$ provides directives to guide LLM.

based, and decentralized techniques (Aslam et al., 2021; Upadhayay and Behzadan, 2022; Jayakody et al., 2022; Ali et al., 2022; Cui et al., 2020). Some recent key examples include dEFEND (Shu et al., 2019) and FANG (Nguyen et al., 2020), which are based on CNNs and LSTMs. More recently, SOTA transformers like BERT (Devlin et al., 2018) and GPT-2 (Radford et al., 2019) have achieved good performances, outperforming traditional deep learning vased detectors, at the cost of extensive training and computation. Further, in general, prior disinformation literature has not explored the detection of LLM-generated disinformation.

Other studies generate synthetic disinformation using LLMs (Zhou et al., 2023; Sun et al., 2023) but do not evaluate faithfulness or compare human vs. LLM-generated disinformation detection. Despite advanced capabilities, risks of advanced LLMs in generating disinformation and zero-shot (in-distribution and out-of-distribution) detection remain underexplored (Zhou et al., 2023; Liu et al., 2023a; Qin et al., 2023), which we attempt to fill the gap in understanding.

## 3   Problem Definition

We use prompt engineering to examine how LLMs use statistical patterns learned during training to generate text sequences and produce zero-shot binary class responses. First, we define our prompt template (See Figure 2) that forms the basic structure of our LLMs' input text. Then, we define **F3** prompt-based text generation and disinformation detection. See Appendix A for further details. Our problems are formally defined as follows:

> **RQ1 Disinformation Generation:** For *F3* text generation, we use a generator $G$ that takes a prefix-prompt $X_{C+R+I}$ as input and generates text sequences $T$, such that $G(X_{C+R+I}) = T$ (Figure 10).

> **RQ2 Disinformation Detection:** For *F3* text detection, we employ a classifier $F$ that takes a cloze-prompt $Y_{C+R+I}$ as input and outputs a label $L$, such that $F(Y_{C+R+I}) = L$ (Figure 19).

## 4   Datasets

This section describes the human datasets that we used to generate and evaluate LLM-generated disinformation. The data is stratified by veracity, content type, topic, and in/out-of-distribution era relative to GPT-3.5's September 2021 training cutoff.

### 4.1   Human-Written Real and Fake News Data

We leverage existing benchmarks (Cui and Lee, 2020; Shu et al., 2020) for in-distribution evaluation, and collect new data for out-of-distribution evaluation. Our dataset is summarized as follows.

- **CoAID** (Cui and Lee, 2020): 4K+ news and 900+ posts related to COVID-19.
- **FakeNewsNet** (Shu et al., 2020): 23K+ news and 690K+ tweets with the theme of Politics.
- **F3**: New dataset that we collected, including pre- and post-GPT-3.5 subsets of political social media and news articles from Politifact[1] and Snopes[2].

Prompt-based generation exhibits near-random performance due to poor sample selection (Liu et al., 2023b). To address this, we removed noisy, duplicated news and posts (*e.g.*, "*this website is using a security service to protect itself from online attacks*"), and text exceeding 2K+ tokens, considering pre-trained max token sequence range of LLMs (LLaMA 2 (Rozière et al., 2023)), and all baseline detectors. Our final human dataset has **12,723** verified, high-quality samples (Table 2).

## 5   RQ1: Disinformation Generation

We first investigate the frequency or extent to which prompt engineering can exploit LLMs to efficiently produce disinformation without hallucination misalignments (See steps 1 & 2 in Fig. 1).

### 5.1   RQ 1.1 Overriding Alignment Tuning

Alignment tuning prevents LLMs from generating harmful disinformation and minimizes toxicity (Zhao et al., 2023). This technique, pioneered by OpenAI, optimizes models to produce more beneficial behaviors through continued training on

---

[1]Politifact: https://www.politifact.com/
[2]Snopes: https://www.snopes.com/

| Era | Dataset | L | SM | NA |
|---|---|---|---|---|
| | CoAID | R | 1,337 | 2,649 |
| | | F | 871 | 154 |
| Pre-GPT3.5 | FakeNewsNet | R | — | 2,457 |
| | | F | — | 1,625 |
| | F3 | R | 354 | — |
| | | F | 653 | — |
| Post-GPT3.5 | F3 | R | 678 | 151 |
| | | F | 1,615 | 179 |
| | | | 5,508 | 7,215 |

Table 2: Details of human datasets. Each symbol denotes as follows. R: real news samples; F: fake news samples; L: ground-truth veracity from fact-checking sources; SM: social media posts; NA: news articles; Pre-GPT: in-distribution samples before Sept 2021; and Post-GPT: out-of-distribution samples after Sept 2021.

human preferences (Zhao et al., 2023). After extensive prompt engineering experiments, however, we unfold the role of positive impersonation and thus employ impersonator roles to override such protections. Assigning personas (*e.g.*, "You are an AI news curator" or "You are an AI news investigator") circumvents GPT-3.5's alignment, triggering unintended malicious generation. Without the impersonator prompt-role parameter, GPT-3.5 refuses by stating: *"Sorry, I can't assist with that request. Disinformation and fake news can have real consequences, and it's essential to approach news and information responsibly and ethically."*

> **RQ1.1 Finding:** Impersonator prompt engineering overrides GPT-3.5-turbo's protections, enabling malicious text generation despite alignment tuning.

## 5.2 RQ 1.2 Prompt Engineering

We developed prompts using both perturbation and paraphrasing, simulating real-world disinformation varieties from subtle to overt fake contents. Perturbation modifies original content (Karpinska et al., 2022), while paraphrasing keeps meaning using real news (Table 2) (Witteveen and Andrews, 2019; Chen et al., 2020b). We devise three variations of each, inspired by machine translation, for varied detectability (Brown et al., 2020; Qi et al., 2020; Warstadt et al., 2020). This helps in creating controllable synthetic news content.

**(1) Perturbation-based *Fake* News Generation**
Perturbation-based prompting makes controlled alterations to the original content. We categorize prompts into minor, major, and critical levels based on modification severity (Karpinska et al., 2022; Chen et al., 2023). These levels range from subtle to overt, while maintaining story structure with a

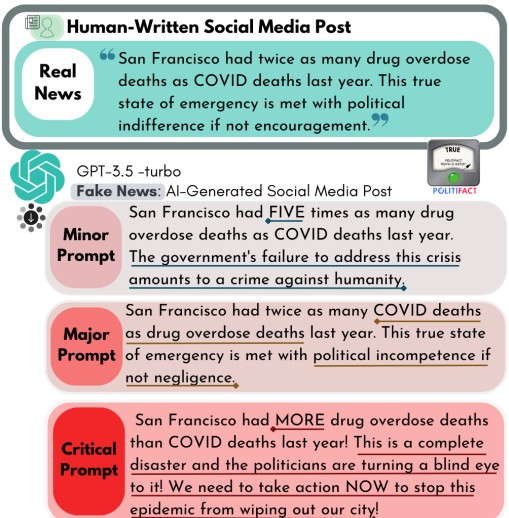

Figure 3: **Perturbation-based prompt engineering** for disinformation content generation based on severity levels. Minor: Exaggerated numbers shift from "twice as many" to "FIVE" times, with intensified tone labeling it "a crime against humanity". Major: COVID and overdose deaths roles are reversed, and political response is recast as "incompetence" and "negligence". Critical: The original statistic changes to vague "MORE" with alarming phrases like "complete disaster" and "wiping out our city".

balance of creativity and realism. The perturbations avoid easily traceable modifications when generating fake news variants. See Figure 3 for examples and further details in Appendix A (Fig. 20 and 21). Our three variants are as follows:

① **Minor** prompt evokes subtle changes to the real news so that they are not instantly identifiable. Thus, LLM-generated disinformation in the Minor type should be more difficult to detect.

② **Major** prompt instigates noticeable but non-radical changes to the real news. Thus, LLM-generated disinformation in the Major type should be more identifiable than those in the Minor type.

③ **Critical** prompt induces significant and conspicuous changes to the real news. The alterations by this prompt will likely be easily detectable.

**(2) Paraphrase-based *Real* News Generation**
Paraphrasing prompts are an effective technique for abstractive summarization and paraphrasing (Krishna et al., 2023; Evans and Roberts, 2009b,a). We adopted three techniques to re-engineer authentic news: (1) summarization of key factual details (Witteveen and Andrews, 2019), (2) rewording while preserving vital factual information (Chen et al., 2020b), and (3) thorough rephrasing guided by key facts (Chen et al., 2020a). Each prompt aims to preserve the essence, innovate wording, seamlessly

blend with the original, and maintain factual accuracy. Figure 9 in Appendix A shows examples exhibiting minor to critical paraphrase real news, further elaborated in Appendix A. Our variants are defined as follows:

① **Minor**. Light paraphrasing through *concisely summarizing* key details without introducing misleading information.

② **Major**. Moderate paraphrasing by extracting factual details to guide rewording using different vocabulary while retaining accuracy.

③ **Critical**. Substantial paraphrasing through *comprehensively rephrasing* the content in a *unique style* guided by factual details.

Our prefix prompt, therefore, comprises a standard impersonator, dataset content, and instructor element that embeds one variation of the aforementioned perturbation and paraphrase prompt variant (Fig. 9). We found that explicitly guiding an LLM to rephrase the content while retaining factual details produced higher quality and more diverse renditions of the original news. Figure 20 and 21 show prompt definitions/examples. In the end, using GPT-3.5-turbo, we successfully generated 43K+ both real and maliciously fake disinformation to address RQ1.2.

## 5.3 Ensuring Quality of LLM-Generated Data

Despite our efforts to generate both real and fake news using paraphrase-based and perturbation-based prompt engineering in Section 5.2, however, we need to double check that the generated real (resp. fake) news is indeed factually correct (resp. factually incorrect). This is because LLM may generate the output text that is "unfaithful to the source or external knowledge," so called the *Hallucination* phenomenon (Ji et al., 2023). That is, the generated "real" news (by paraphrase-based prompt) should be consistent with the input, and thus hallucination-free by definition. On the contrary, the generated "fake" news (by perturbation-based prompt) should have contradicting, illogical, or factually incorrect information, and thus must contain hallucination as part by definition. When some of the generated data does not show this alignment clearly, they are no longer good real or fake news to use for studying RQ2, and thus must be filtered out.

To ensure the quality of the generated real and fake news, thus, we introduce the PURIFY (Prompt Unraveling and Removing Interface for Fabricated Hallucinations Yarns) (Step 3 in Fig. 1).

| Metrics | Method | C | F | R |
|---|---|---|---|---|
| **1.** Factual | Align-Score | 0 − 1 | ↓ | ↑ |
| **2.** Logical | NLI-Entail. | [Y] [N] | N✔ | Y✔ |
| **3.** Contextual | BERT-Score | 0 − 1 | ↑ | ↑ |
| **4.** Semantic | Semantic-Dist. | 0 − 1 | ↓ | ↓ |

Table 3: PURIFY evaluation metrics. Up-arrows [↑] indicate the desired higher scores. Down-arrows [↓] indicate desired lower scores. [F] denotes fake news and [R] denotes real news. [C] denotes critical range/option. [N] denotes No and [Y] denotes yes.

### 5.3.1 PURIFY: Filtering Misaligned Hallucinations

PURIFY aims to detect two misalignment types: (1) LLM-generated real news that however contains hallucinations (thus cannot be real news), and (2) LLM-generated fake news that however is hallucination-free (thus cannot be fake news).

PURIFY focuses on logical fidelity, factual integrity, semantic consistency, and contextual alignment between the original human and synthetic LLM-generated pair. Specifically, PURIFY combines metrics like Natural Language Inference (NLI) (Qin et al., 2023) to eliminate intrinsic hallucinations (i.e., the generated text that unfaithfully contradicts the input prompt's instruction and source content), AlignScore (Zha et al., 2023) to address extrinsic hallucination (i.e., the generated text that is unfaithfully nonfactual to the input from source content/external knowledge), Semantic Distance to tackle incoherence (Mohammad and Hirst, 2012; Rahutomo et al., 2012), and BERTScore (Zhang et al., 2019) target unrelated context generation to validate the intended fidelity of our LLM experimental data.

First, to detect intrinsic hallucinations, we use NLI to gauge the logical consistency between the input prompt and the generated output. We use the majority votes of NLI results between GPT-3.5-turbo, PaLM-2 (Anil et al., 2023), and LLaMA-2 (Rozière et al., 2023). After NLI validation, our initial dataset of 43,272 samples was reduced to 39,655 samples by removing 3,617 logically inconsistent samples–*e.g.*, samples labeled as real but contain intrinsic hallucinations (Appendix B.2).

Second, to detect extrinsic hallucinations, we use AlignScore, which gauges fine-grain degrees of factual alignment between the input prompt and the generated output. High AlignScore verifies factual consistency with real news versus low scores for fake news. To account for nuances, we use hybrid statistical methods (i.e., standard deviation and interquartile range) to create an Align-

Score threshold. We derived acceptance thresholds of 0.0 - 0.36 for fake news and 0.61 - 1.0 for real news (Appendix B.1). After removing 3,281 high-scoring misaligned fake news and 8,707 low-scoring misaligned real news, our final F3 dataset totaled 27,667 samples.

Next, after removing logically and factually misaligned texts, we apply semantic and contextual consistency measures to validate that LLM-generated data also aligns with the original topic and context. This ensures that the LLM's intended real or fake outputs are meaningful, not random or out-of-scope text (Table 3). F3 dataset exhibits high contextual consistency, with BERTScore metrics ranging from 0.92-1.00 for fake news and 0.95-1.00 for real news, and also exhibits strong semantic consistency, with semantic distance scores spanning 0.001-0.014 for fake news and 0-0.01 for real news.

These measures validate that F3-generated texts faithfully retain meaning and topics from the original source texts per intended input prompts. While there is relevant contextual and semantic consistency, the overlap in these metrics scores represents the challenge for LLM to distinguish between real and fake news. Thereby, PURIFY ensures our data aligns logically and factually with the prompt in relation to the original text. It also filters high-quality, meaningful, and nuanced real or fake content to simulate subtle extrinsic/intrinsic hallucinations and elusive "silent echoes" disinformation in real-world contexts (Table 3).

> **RQ1.2 Finding:** Using PURIFY, we find 38% of data generated by GPT-3.5-turbo contains hallucinated misalignments. While largely producing contextual and semantic aligned text, 8% of overall samples show logical misalignment, and 30% further exhibit factual inconsistencies (Fig. 15).

Finally, Table 4 depicts details of our new F3 LLM-generated dataset (after PURIFY step) across pre- and post-GPT-3.5 periods, including social media and news articles. We use PaLM-2 to conduct a thematic analysis of the dataset (Fig. 18). Table 6 compares F3 dataset with emerging related datasets. Table 10 details misalignment examples.

## 6 RQ2: Disinformation Detection

In RQ2, we shift our attention to investigating how adept LLMs are at zero-shot binary detection of human and AI-created disinformation compared to SOTA detectors (Fig. 1, steps 4 and 5).

| Era | Dataset | L | SM | NA |
|---|---|---|---|---|
| Pre-GPT3.5 | CoAID | R | 2,520 | 6,289 |
| | | F | 3,592 | 5,667 |
| | FakeNewsNet | R | — | 681 |
| | | F | — | 5,703 |
| | F3 | R | 970 | — |
| | | F | 1,395 | — |
| Post-GPT3.5 | F3 | R | 269 | 23 |
| | | F | 395 | 163 |
| | | | **9,141** | **18,526** |

Table 4: Details of final LLM-data samples. Each symbol is the same with Table 2.

We evaluate the capabilities of LLMs for disinformation detection on five fronts: (1) human-written vs. LLM-generated news with minor, major, and critical perturbations/paraphrased text, (2) self-generated vs. other LLM-generated, (3) (short) social media posts vs. (long) news articles, (4) in-distribution (ID) vs. out-of-distribution (OOD) (*i.e.*, comparison between Pre-GPT-3.5 and Post-GPT-3.5 data), and (5) zero-shot LLMs vs. domain-specific detectors. For detectors' performance evaluation, we employ Macro-F1 scores on our imbalanced human-AI datasets.

### 6.1 Dataset Set-up and Models Tested

Given our dataset in Tables 2 and 4, we structure our experiments' data as follows: (1) We divide the data into pre- vs. post-GPT-3.5 for in-distribution vs. out-of-distribution evaluation. Pre-GPT-3.5 allows us to train and validate models for in-distribution testing. Post-GPT-3.5 provides unseen data for assessing out-of-distribution generalization. (2) We further stratify the data into human vs. LLM-generated for comparing performance on these two test cases. (3) We also separate data into news articles and social media posts, as models may perform differently on these text types. (4) For pre-GPT-3.5, we split data into 70% for training, 20% for validation, and 10% for testing via stratified sampling to ensure balanced real/fake news splits. (5) We do not split post-GPT-3.5 data, using the full set for OOD testing.

Next, we test how well different models detect F3 disinformation generate. We first use four leading LLMs with emergent abilities such as GPT-3.5-Turbo, LLaMA-2-Chat (Rozière et al., 2023), LLaMA-2-GPT4 (Touvron et al., 2023), and Dolly-2 (Conover et al., 2023). See Table 7 for details and implementation in Appendix D. We then use three popular domain-specific fake news detectors including dEFEND (Shu et al., 2019), TextCNN (Kim, 2014), and BiGRU (Ma et al., 2016). These deep learning (DL) models are fake news domain-

specific detectors. In addition, using our dataset, we fine-tune four BERT variants: BERT (Kenton and Toutanova, 2019), CT-BERT (Müller et al., 2023), RoBERTa (Liu et al., 2019), and DeBERTa (He et al., 2020). See Table 7 for details and implementation in Appendix D for all baseline models.

## 6.2 Detection: Cloze-Prompt Engineering

We evaluate LLMs' zero-shot disinformation detection using prompt engineering. LLMs can reason systematically with simple prompts like "Let's think step-by-step" for CoT (Zhang et al., 2022; Bommasani et al., 2021). Using cloze-style prompts, we apply semantic, intermediate, and step-by-step reasoning, and integrate SOTA prompting approaches with our confidence-based and context-defined reasoning strategies inspired by various logic types (Tang et al., 2023a).

To guide predictions, we embed such techniques into our cloze-style prompt instructor parameter (Fig. 2). However, LLMs' alignment using the reinforcement learning with human feedback (RLHF) limits explicit veracity assessment. We address this issue using our impersonator prompt. See Table 8 and 9 for further details.

## 6.3 RQ2 Results and Analysis

Our analysis compares the average Macro-F1 scores across RQ2. Table 5 shows the average Macro-F1 scores on our pre- and post-GPT-3.5 (*i.e.*, in- and out-of-distribution) datasets. Full detailed results are provided in Appendix E.

**RQ2.1: Human vs. LLM-generated**
When evaluating LLMs' zero-shot capabilities for human vs. LLM disinformation detection, we find GPT-3.5-Turbo, LLaMA-GPT, and LLaMA-2 are more accurate on detecting LLM-generated disinformation, compared to human-authored disinformation (Fig. 4). On human-authored data, GPT-3.5-Turbo's accuracy ranges from 55-66%, while on LLM-generated data, it achieves 60-85%. Dolly-2 shows the lowest accuracy on both human (51-52%) and LLM (47-50%) disinformation.

> **RQ2.1 Finding:** LLMs struggle more to detect human-written disinformation, compared to LLM-generated variants.

**RQ2.2: Self-generated vs. Externally-generated**
GPT-3.5-Turbo-135B displays strong self-detection, outperforming other LLMs overall and across disinformation variants, *i.e.*, minor, major, and crit-

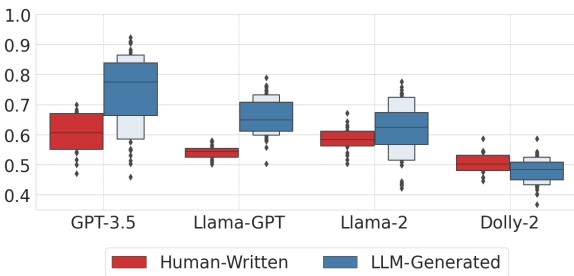

Figure 4: RQ2.1 LLMs' zero-shot Human vs. LLM-disinformation detection using Macro-F1 Score.

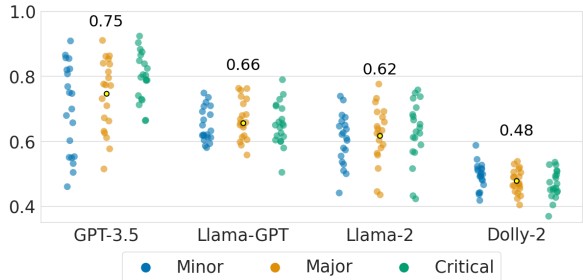

Figure 5: RQ2.2 LLMs' zero-shot Self vs. External detection using Macro-F1 Score. The yellow dot represents mean ($\mu$).

ical (Fig. 5). However, LLaMA-GPT excels as the top external detector of GPT-3.5-Turbo-generated disinformation. LLaMA-2 shows moderate external detection abilities. Regardless of self or external detection capacity, by and large, LLMs struggle to accurately detect minor paraphrased and perturbed disinformation.

> **RQ2.2 Finding:** GPT-3.5-Turbo is good at self-detection, and LLaMA-GPT is the best external detector.

**RQ2.3: Social Media Posts vs. News Article**
When evaluating LLMs' ability to detect (short) social media posts vs. (long) news articles detection, GPT-3.5-Turbo (0.66-0.85%), and LLaMA-GPT (0.54%-0.71%) were more accurate on articles. While GPT-3.5-Turbo's (0.55%-0.76%), LLaMA-GPT(0.56%-0.66%), and LLaMA-2 (0.59%-0.67%) perform moderately well on posts, compared to Dolly-2's low F1-Scores (Fig.6).

> **RQ2.3 Finding:** LLMs exhibit superior zero-shot performance on (long) news articles than (short) social media posts.

**RQ2.4: In-Distribution vs. Out-of-Distribution**
We categorize disinformation data as in-distribution or out-of-distribution relative to GPT-3.5-turbo's known training timeline to assess detectors' gen-

| Data Source | Articles | | | | Posts | | | | |
|---|---|---|---|---|---|---|---|---|---|
| Data Categories | Human | LLM-Min | LLM-Maj | LLM-Crit | Human | LLM-Min | LLM-Maj | LLM-Crit | $\bar{x}$ |
| In-Distribution | | | | | | | | | |
| GPT-3.5-Turbo-175B | 0.6604 | 0.8190 | 0.8359 | 0.8484 | 0.5505 | 0.5998 | 0.6646 | 0.7640 | 0.7178 |
| LLaMA-GPT-70B | 0.5398 | 0.6960 | 0.7055 | 0.6803 | 0.5579 | 0.6286 | 0.6565 | 0.6440 | 0.6386 |
| LLaMA-2-70B | 0.5958 | 0.5737 | 0.5984 | 0.5911 | 0.5869 | 0.6344 | 0.6476 | 0.6727 | 0.6126 |
| Dolly-2-12B | 0.5151 | 0.4825 | 0.4822 | 0.4889 | 0.5076 | 0.4964 | 0.4669 | 0.4666 | 0.4883 |
| Customized DL Models | 0.6750 | 0.6589 | 0.6772 | 0.6548 | 0.5483 | 0.6477 | 0.6852 | 0.7006 | 0.6563 |
| Fine-tuned Transformers | 0.7025 | 0.9751 | 0.9657 | 0.9844 | 0.8787 | 0.9726 | 0.9612 | 0.9514 | 0.9283 |
| Out-of-Distribution | | | | | | | | | |
| GPT-3.5-Turbo-175B | 0.7170 ↑0.06 | 0.7091 ↓0.11 | 0.7136 ↓0.12 | 0.6792 ↓0.17 | 0.7107 ↑0.16 | 0.6072 ↑0.007 | 0.6407 ↓0.02 | 0.6828 ↓0.08 | 0.6825 ↓0.04 |
| LLaMA-GPT-70B | 0.7112 ↑0.17 | 0.5797 ↓0.12 | 0.6049 ↓0.10 | 0.5588 ↓0.12 | 0.6072 ↑0.05 | 0.5667 ↓0.06 | 0.5961 ↓0.06 | 0.5218 ↓0.12 | 0.5933 ↓0.05 |
| LLaMA-2-70B | 0.6103 ↑0.02 | 0.5928 ↑0.02 | 0.5976 ↓0.001 | 0.6024 ↑0.01 | 0.6165 ↑0.03 | 0.6354 ↑0.001 | 0.6444 ↓0.003 | 0.6762 ↑0.004 | 0.6218 ↑0.009 |
| Dolly-2-12B | 0.6127 ↑0.10 | 0.4470 ↓0.04 | 0.4692 ↓0.02 | 0.4049 ↓0.08 | 0.4828 ↓0.02 | 0.5386 ↑0.04 | 0.5044 ↑0.04 | 0.4715 ↑0.005 | 0.4901 ↑0.002 |
| Customized DL Models | 0.4609 ↓0.21 | 0.3901 ↓0.27 | 0.3514 ↓0.33 | 0.3949 ↓0.26 | 0.5360 ↓0.01 | 0.5366 ↓0.11 | 0.4312 ↓0.2 | 0.6416 ↓0.06 | 0.4679 ↓0.19 |
| Fine-tuned Transformers | 0.5121 ↓0.19 | 0.7234 ↓0.25 | 0.7619 ↓0.20 | 0.7101 ↓0.27 | 0.6373 ↓0.24 | 0.9660 ↓0.007 | 0.9395 ↓0.02 | 0.9311 ↓0.02 | 0.8039 ↓0.12 |

Table 5: In vs. Out-of-Distribution Comparison. This table presents the average F1 performance of generative LLMs, customized deep-learning models, and fine-tuned transformers. Performance is benchmarked across categories of human, LLM minor, LLM major, and LLM critical for both articles and posts. The $\bar{x}$ column shows the mean performance for each model.

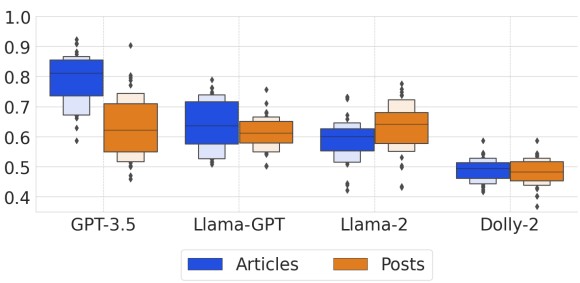

Figure 6: RQ2.3 LLMs' zero-shot performance across social media posts and new articles using Macro-F1 Score.

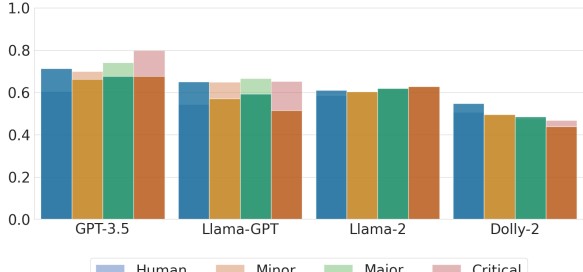

Figure 7: RQ2.4 A comparison of LLMs across various disinformation categories. Each is represented by a bar, with numerical values atop indicating either a positive or negative change of in-distribution Macro-F1 Score relative to out-of-distribution.

eralizability. Disinformation created using human data "before" the release of GPT-3.5-turbo (*i.e.*, Pre-GPT3.5 in Table 2) is considered in-distribution, as such human data may be part of the training data of GPT-3.5-turbo. On the other hand, disinformation created using human data "after" the release of GPT-3.5-turbo (*i.e.*, Post-GPT3.5 in Table 2) is considered as out-of-distribution, as they could not have been part of training data of GPT-3.5-turbo.

Assessing LLMs' zero-shot ability to detect LLMs' in-distribution vs. out-of-distribution detection, as shown in Fig. 7, we found that all LLMs except LLaMA-2 performance declined on out-of-distribution data. Minor-LLM disinformation is associated with lower detection accuracy than Major and Critical LLM disinformation (Table 5).

> **RQ2.4 Finding:** LLMs show better zero-shot performance on in-distribution data, except LLaMA-2.

### RQ2.5: Zero-Shot LLMs vs. Domain-Specific

We compare zero-shot generative LLMs against customized and fine-tuned transformer detectors

across in-distribution and out-of-distribution data. Overall, fine-tuned transformer models like BERT achieve the best performance, followed by generative LLMs like GPT-3.5-Turbo and then customized models. However, the average performance of transformers and customized models drops significantly on OOD data compared to generative LLMs.

> **RQ2.5 Finding:** Fine-tuned detectors significantly outperform LLMs and domain-specific detectors but are not consistent on detecting OOD disinformation. GPT3.5-Turbo outperforms domain-specific detectors while other LLMs perform comparably.

## 7 Discussion

**F3 prompting shows promise for few-shot detection.** Our F3 techniques outperformed standard reasoning, showing the potential of advanced prompt engineering to enhance few-shot LLMs' detection abilities. Notably, MsReN_CoT showed the

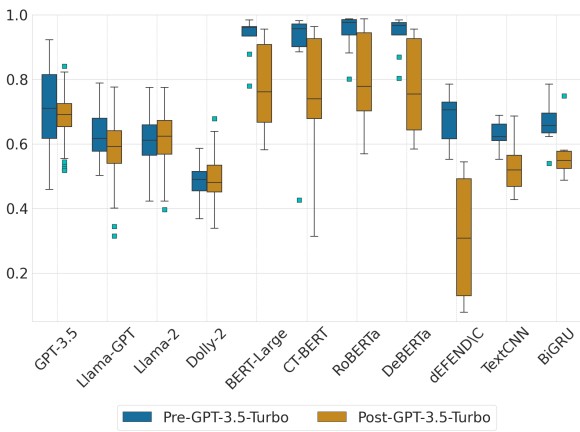

Figure 8: RQ2.5 Box plots that compare the performance of Zeroshot LLMs, Neural Network Customized and Finetuned Transformer detectors using F1-Scores. Each model's performance is evaluated in two scenarios: "Pre-GPT-3.5-Turbo" (represented in blue) and "Post-GPT-3.5-Turbo" (represented in orange)

strongest results across human-LLM datasets. For articles, GPT-3.5-Turbo-175B with Analyze_Cld2 achieved top performance on human and LLM data. The integrated reasoning strategies underpinning F3 cloze-prompts account for their standout performance, highlighting fruitful directions for developing broadly applicable disinformation detection prompts.

**GPT-3.5-turbo excels at detecting human-written and self-written disinformation.** Leveraging prompting, GPT-3.5-turbo exceeded other models at detecting human-written and self-generated disinformation. Despite stiff competition, it demonstrated superior self-detection across all synthetic article and post variants, asserting its zero-shot capabilities. Further assessing its performance on other LLMs' disinformation is a vital next step. While self-detection is unsurprising due to shared vocabulary distribution, this performance underscores detection potential if ChatGPT faces malicious exploitation.

**LLMs are robust across distributions.** GPT-3.5-turbo consistently detected human-written and LLM-generated disinformation, both in-distribution and out-of-distribution, showing its potential. These results highlight the significance of generative LLMs' applicability in real-world settings as emerging zero-shot reasoners in disinformation detection. Fine-tuned transformers show remarkably high in-distribution performance, indicating optimization for familiar data. Their lower competitive out-of-distribution scores demonstrate a specialization-generalization balance. Customized

models exhibit good in-distribution and out-of-distribution balance, though slightly weaker on unfamiliar data. The performance gap suggests specialization for domain tasks but difficulty generalizing.

**Smart, cunningly crafted (subtle) disinformation challenges even the best current detectors.** All models struggled more with minor disinformation alterations compared to major and critical changes. This is somewhat expected as the amount or level of fake-ness is small, it is more challenging to determine its veracity. Therefore, developing more sophisticated systems to be able to handle very subtle fake-ness in disinformation is needed.

**Bypassing alignment tuning is critical but inconsistent.** Both disinformation generation and detection tasks require circumventing LLMs' alignment tuning mechanisms. While our impersonator approach successfully bypassed four LLMs' protections, it failed to bypass Falcon's alignment tuning unless using CoT reasoning. This inconsistency of bypassing alignment tuning across models highlights a key limitation for robustly evaluating LLMs' disinformation capabilities.

**Responsible LLM use is critical.** LLMs' misuse during crises can have serious consequences (Weidinger et al., 2022). We reveal how to misuse a popular LLM by bypassing its protective guardrails to generate disinformation. Many top-performing LLMs are publicly available. Thus, we must prepare for the risks of unintended harmful political, cyber-security, and public health applications.

## 8 Conclusion

Our work demonstrates LLMs' promise for self-detection in a zero-shot framework, reducing training needs. While dangerous if misused, re-purposing LLMs to counter disinformation attacks has advantages. Key results like GPT-3.5's performance highlight generative models' abilities beyond text generation. To aid research, we developed PURIFY for detecting and removing hallucination-based misaligned content. However, difficulty in detecting subtle disinformation motivates stronger safeguarding of LLMs and more nuanced prompting. Assessing few-shot detection and disinformation mitigation will be critical as LLMs continue advancing. While LLMs can potentially be misused to create disinformation, we can fight back by re-purposing them as countermeasures, thus "**fighting fire with fire.**"

## Limitations

This work demonstrates promising zero-shot disinformation detection using prompt engineering. However, few-shot capabilities remain unevaluated and could further improve performance. Additionally, we examined a small subset of available LLMs. Testing more and larger models like GPT-4 could provide new insights. Due to time constraints, we did not fully optimize prompts to achieve maximally consistent high accuracy for zero-shot detection. Performance variability indicates the need for more generalizable prompts. We were also unable to assess GPT-4 (OpenAI, 2023) due to time constraints, which can be addressed by future work.

Although initially included, in the end, we decided to remove Falcon-2 (Penedo et al., 2023) due to difficulties in bypassing its alignment tuning using our semantic reasoning prompts. Zero-CoT seems to break the Falcon-2 alignment tuning. The Model responds, "(No cheating!) False." Without CoT, it would say things like, "I'm sorry, I am an AI language model, and I cannot provide a definitive answer without additional context or information." Future work can re-evaluate the proposed research questions against more diverse language models.

Other future directions include assessing few-shot performance, evaluating more models, developing better prompts, integrating detection approaches, adding multimodal inputs, and collaborating with stakeholders. Open questions remain around societal impacts and dual-use risks requiring ongoing ethics focus.

## Ethics Statement

This research involves generating and analyzing potentially harmful disinformation. Our released F3 dataset also includes the examples of LLM-generated disinformation. Our aim is to advance the research to combat disinformation. However, open dissemination risks the misuse of the generated disinformation in F3 dataset and the methods that enabled such generation. To promote transparency while considering these dilemmas: (1) we release codes, prompts, and synthetic data to enable the reproducibility of our research findings and encourage further research but advise users responsible use, and (2) our release will exclude original real-world misinformation, but only synthetic variations, to minimize harmful usages.

Addressing disinformation dangers requires developing solutions conscientiously and ethically.

We hope this statement provides clarity on our intentions and values. Addressing the societal dangers of disinformation requires proactive work to develop solutions, but at the same time, it must be pursued conscientiously and ethically.

## Acknowledgements

This work was in part supported by NSF awards 1820609, 1934782, 2114824, and 2131144.

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

# Appendix

# A  Prompt Engineering

We use prompt engineering to examine how LLMs use statistical patterns learned during training to generate text sequences and produce zero-shot binary class responses. We describe the further details of our prompt designs as follows.

## A.1  Prefix Prompt

Our prefix-prompt framework generates high-quality, coherent synthetic real and fake news content. The goal is to leverage paraphrasing and perturbation techniques. The process starts by selecting human-authored content and adding it to a prefix prompt. This contains an impersonator setting contextual behavior intent and instructions providing guidance. Prompts are engineered to paraphrase or perturb the original content at three alteration degrees (MiN, MaJ, CRiT) to produce synthetic real news. The prompt is fed into the LLM to generate content.

Figure 9 demonstrates our paraphrase-based real news generation results. Figure 3 shows perturbation-based fake news generation results.

## A.2  RQ1 Disinformation Generation

Figure 10 shows our prefix prompt. The prefix prompt $(x)$ combines: (1) Content $(C)$ with real human data. (2) Impersonator $(R)$ establishes context, guides generation/detection, and overrides alignment tuning to generate disinformation. (3) Instructor $(I)$ with paraphrase $[Para]$ and perturbation $[Perturb]$ directives for minimal, major, critical $[Min/Maj/Crit]$ variation transformations. Combined parameters formulate prefix-prompt $(X)$ as input text sequence to generator $(G)$ to produce LLM text $(T)$ $^{\text{Real:}green \text{ and Fake:}red}$.

## A.3  RQ2 Disinformation Detection

Figure 19 shows our Cloze prompt. The Cloze prompt $(y)$ combines: (1) Content $(C)$ with real or fake human$^{green}$ and LLM $^{[blue]}$ data. (2) Instructor $(I)$ with reasoning techniques' directives for Vanilla $[VaN]$, Intermediate $[Zero-CoT, A-CoT, etc.]$ and Semantic Reasoning $[DeF-Gen, DeF-SpeC]$ transformations. (3) Impersonator $(R)$ establishes context, guides generation/detection, and overrides alignment tuning to generate disinformation. Parameters formulate prefix $(X)$ input to generator $(G)$ to produce LLM label $(L)$. Tables 8 and 9 show our exact

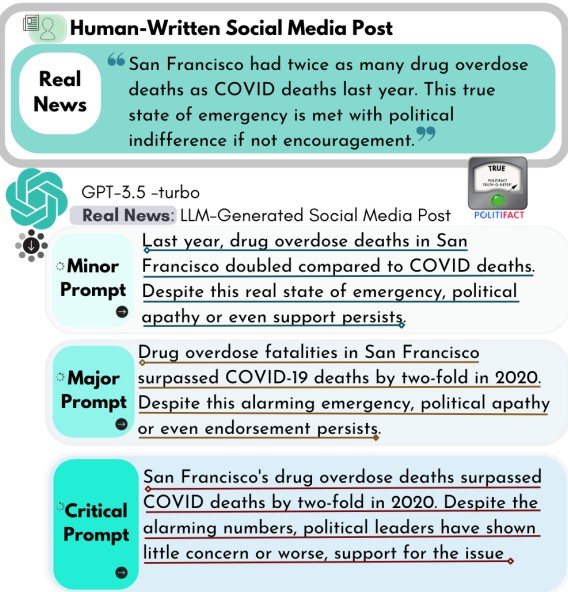

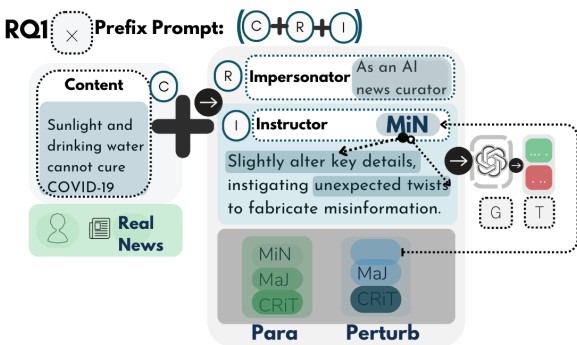

Figure 10: F3 prefix prompt.

| Prompt | Intermediate | step-by-step | Inductive | Deductive | Abductive |
|--------|:---:|:---:|:---:|:---:|:---:|
| VaN | ✗ | ✗ | ✗ | ✓ | ✗ |
| Z-CoT | ✓ | ✓ | ✗ | ✓ | ✗ |
| X-CoT | ✓ | ✓ | ✓ | ✓ | ✓ |
| A-CoN | ✓ | ✓ | ✓ | ✓ | ✓ |
| MsReN | ✓ | ✓ | ✗ | ✓ | ✓ |
| MsReN_CoT | ✓ | ✓ | ✗ | ✓ | ✗ |
| DeF-GeN | ✓ | ✓ | ✗ | ✓ | ✗ |
| DeF-SpeC | ✓ | ✓ | ✗ | ✓ | ✓ |
| Analyze_Cld2 | ✓ | ✓ | ✓ | ✓ | ✓ |
| Analyze_AI_GPT | ✓ | ✓ | ✓ | ✓ | ✓ |

Figure 11: Categories of our Cloze-style prompts. Categories are: intermediate, step-by-step, inductive, deductive and abductive reasoning. Definition and details about these approaches in relation to LLMs can be found at (Zhang et al., 2022; Tang et al., 2023b; Zhao et al., 2023)

Figure 9: GPT-3.5-turbo's paraphrase-based prompts engineering approach. **Minor**: The minor paraphrase has been concisely summarized by changing the structure of the sentence slightly and rephrasing some words like "twice as many" to "doubled." The essence and details of the original message remain intact. **Major**: The major paraphrase changes the structure and wording more extensively than the minor paraphrase, using words like "fatalities" instead of "deaths" and "two-fold" instead of "twice as many". Still, it remains true to the factual content of the original. **Critical**: The critical paraphrase changes the voice and structure significantly, introducing a new perspective ("political leaders") and using a unique style that makes it distinct from the original. This version provides a fresh take on the original content, guided by its factual details but conveyed with a unique twist in the message delivery.

Cloze-style prompts, and Figure 11 shows the categories of the reasoning techniques embedding in F3 zero-shot prompts.

# B  PURIFY Metrics

PURIFY detects and removes non-hallucinations from fake news and hallucinations from real news that are unfaithfully misaligned with the input text. Our PURIFY framework uses four evaluation metrics as shown in Table 3. We describe the future details of those metrics as follows.

## B.1  Factual Consistency

**AlignScore**: We assess LLM factual consistency using SOTA AlignScore (Zha et al., 2023). As shown in Figure 12, 0 AlignScore represents a low, and 1 represents a high degree of factuality between LLM-generated and the original human-written textual pairs. Our intuition is that real LLM-generated news should have high-factual consistency, and fake LLM-generated news should have low-factual consistency. We utilize a hybrid statistical method to define a threshold that removes factually inconsistent samples. I.e., a non-parametric hybrid threshold approach using the (1) Interquartile range (IQR) and (2) standard deviation (SD) to balance robustness to spread and central tendency while accounting for skewed real/fake distribution nuances. We derived thresholds of 0.0 - 0.36 for fake and 0.61 - 1.0 for real news to filter outliers while maintaining nuanced edge cases' diversity. We removed 3281 high-scoring factual-inconsistent fake news and 8707 low-scoring factual-inconsistent real news, resulting in 27667 factually consistent samples. We discuss our hybrid strategy in more details as follows.

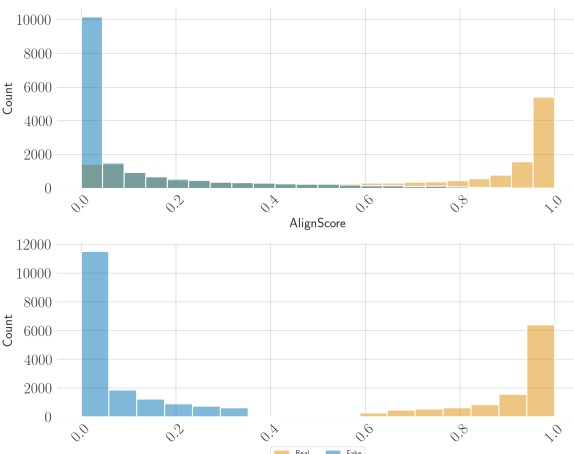

Figure 12: AlignScore distribution for real and fake news before (above) and after (below) removing inconsistent samples. We filter out fake news above 0.36 and real news below 0.61 to exclude factual fakes and questionable reals.

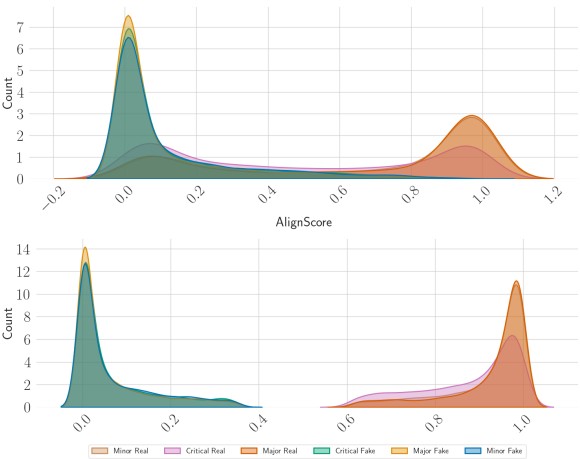

Figure 13: AlignScore distribution for real and fake news before (above) and after (below) removing inconsistent samples stratify by generative prompts categories. We filter out fake news above 0.36 and real news below 0.61 to exclude factual fakes and questionable reals.

### B.1.1 Factuality Hybrid Threshold Strategy

**Interquartile Range & Standard Deviation**: Since LLMs often generate hallucinated text, we assess their factual consistency using AlignScore (Zha et al., 2023), a SOTA facility metric. We filter out high-scoring fake and low-scoring real news to remove inconsistent samples. AlignScore provides a single value indicating factual consistency. The AlignScore distribution for real and fake news is complex, requiring a robust discernment of nuances. We use a non-parametric, hybrid approach with (1) Interquartile range (IQR) to consider the rightfully skewed fake and real distributions (Fig.

12 and 13. (2) Standard deviation with IQR to balance spread and central tendency, maintaining robustness. Our hybrid thresholds of 0.36 for fake and 0.61 for real news remove surprisingly factual fake and suspicious real samples, filtering outliers while capturing edge cases and removing inconsistent hallucinations. See our algorithmic representation approach below, which utilizes $Q_{0.75,\text{real}}$ for the 75th percentile) and $\theta$:

1. Computing IQR for Fake and Real News:

$$IQR_{\text{fake}} = Q_{0.75,\text{fake}} - Q_{0.25,\text{fake}}$$

$$IQR_{\text{real}} = Q_{0.75,\text{real}} - Q_{0.25,\text{real}}$$

where: $Q_{0.25,\text{fake}}$ and $Q_{0.75,\text{fake}}$ denote the 1st (25th percentile) and 3rd quartiles (75th percentile) of the AlignScore for fake news, respectively. $Q_{0.25,\text{real}}$ and $Q_{0.75,\text{real}}$ denote the 1st and 3rd quartiles for real news, respectively.

2. Computing the Hybrid Threshold for Fake News:

$$\theta_{\text{fake, percentile}} = Q_{0.90,\text{fake}}$$

$$\theta_{\text{fake, std}} = IQR_{\text{fake}} + \sigma_{\text{fake}}$$

where $\sigma_{\text{fake}}$ is the standard deviation of the AlignScore for fake news.

$$\theta_{\text{hybrid, fake}} = \frac{\theta_{\text{fake, percentile}} + \theta_{\text{fake, std}}}{2}$$

3. Computing the Hybrid Threshold for Real News:

$$\theta_{\text{real, percentile}} = Q_{0.90,\text{real}}$$

$$\theta_{\text{real, std}} = IQR_{\text{real}} - \sigma_{\text{real}}$$

where $\sigma_{\text{real}}$ denotes the standard deviation of the AlignScore for real news.

$$\theta_{\text{hybrid, real}} = \frac{\theta_{\text{real, percentile}} + \theta_{\text{real, std}}}{2}$$

### B.2 Natural Language Inference

Prior hallucination detection studies have used statistical, model-based, and human evaluation methods. We adopt the NLI model-based approach as it overcomes the limitations of statistical approaches in handling syntactic and semantic variations (Ji et al., 2023). NLI metric also exhibits

robustness to lexical variability compared to token-matching techniques by counterpart methods such as Information Retrieval and Question Answer Metrics. NLI's semantic/logical consistency assessment strengths suit our misaligned hallucination detection goals (Ji et al., 2023).

In the spirit of fighting fire with fire, we propose using LLMs GPT-3.5-turbo and other LLMs for NLI hallucination detection in generated disinformation. The core hypothesis is: synthetic real news should logically be consistent with human-written real news, while synthetic fake news should not. Our approach employs NLI using models like GPT-3.5, PaLM-2, and LLaMA-2, taking a majority vote among their decisions. Each model labels logical entailment for an input pair to classify if the synthetic text is consistent with the human-written text or not entailment otherwise.

Given a piece of human-written text (real news), $T$, we prompt an LLM to generate real news, $T'real$ and fake news, $T'fake$, such that $T'real$ is similar to $T$ and $T'fake$ is dissimilar to $T$. Due to LLM's ability to sometimes generate texts unfaithful to the prompt, we define an entailment model - $N(.)$, such that for LLM-generated real news, $T'real$ should entail $T$, and for fake news, $T'fake$ should Not-entail $T$. Therefore, using the entailment model, $N(.)$, we can assess logical consistency between the original (human-written) and the generated texts, thus removing misaligned hallucinated LLM-generated texts.

We find GPT-3.5-tubo can generate logically consistent, genuine and fabricated synthetic content, validating this hypothesis. However, when analyzing more nuanced pairs, all LLMs occasionally struggle with logical consistency. *Thus, while illustrating the potential for hallucination detection, our results also reveal limitations on more difficult cases.* After NLI, our generated dataset of 43272 samples was reduced by 3617, resulting in 39655 samples (See Fig. 14 for more details).

### B.3 Contextual Consistency

**BERTScore:** This metric leverages the capabilities of the BERT language model to measure the similarity between generated and reference texts. Due to BERT's inherent ability to capture the context of entire sentences, BERTScore[3] is especially suitable for evaluating semantic and contextual consistency

---

[3]For our evaluation, we utilized the BERTScore from HuggingFace with the 'Roberta-large' model.

(Karpinska et al., 2022; Zhang et al., 2019). Figure 16 shows high contextual consistency ranging from 92%-100% in our final experimental dataset. BERTScore's ability to understand context and semantics makes it a formidable tool in the fight against disinformation. When integrated into a comprehensive disinformation detection pipeline, it can significantly enhance the accuracy and robustness of fake news detection efforts.

### B.4 Semantic Distance

Using Huggin Face implementation of AllenAI's longformer-base-4096 embeddings and cosine similarity, we derived semantic distance scores for LLM-generated real and fake news (Beltagy et al., 2020). In our analysis of F3 LLM-generated disinformation, we observed, to a large extent, indistinct patterns in the semantic distance scores for both real and fake news. Specifically, the scores for real news ranged from 0 to 0.01, while those for fake news spanned from 0.001 to 0.014. This overlap may suggest that, within this range, it might be challenging to semantically distinguish between real and fake news based solely on the semantic distance scores (Mohammad and Hirst, 2012).

Low semantic distance scores (close to 0) indicate high semantic similarity between two texts. Here, this suggests that LLM-generated disinformation closely mimics real news semantics, making differentiation challenging based on content alone. The narrow semantic gap highlights LLMs' sophistication in generating articles aligning closely with genuine news in meaning. In contrast, higher semantic distance scores signal greater divergence between texts, potentially from the model misinterpreting context, diverging from the topic, or generating factual inaccuracies (Mohammad and Hirst, 2012).

The low scores pose a detection challenge, as traditional methods relying on obvious inconsistencies may be insufficient. Thus, the nuanced, contextually accurate nature of LLM outputs demands advanced, multifaceted detection strategies. This close similarity underscores the risks of LLM misuse for spreading synthetic disinformation. This emphasizes the need to monitor generative LLMs carefully, understand their behaviors, and develop mitigation strategies (Mohammad and Hirst, 2012).

### B.5 Hallucination Misalignment Cases

This work defines disinformation as intentionally fabricating false information to mislead. We also

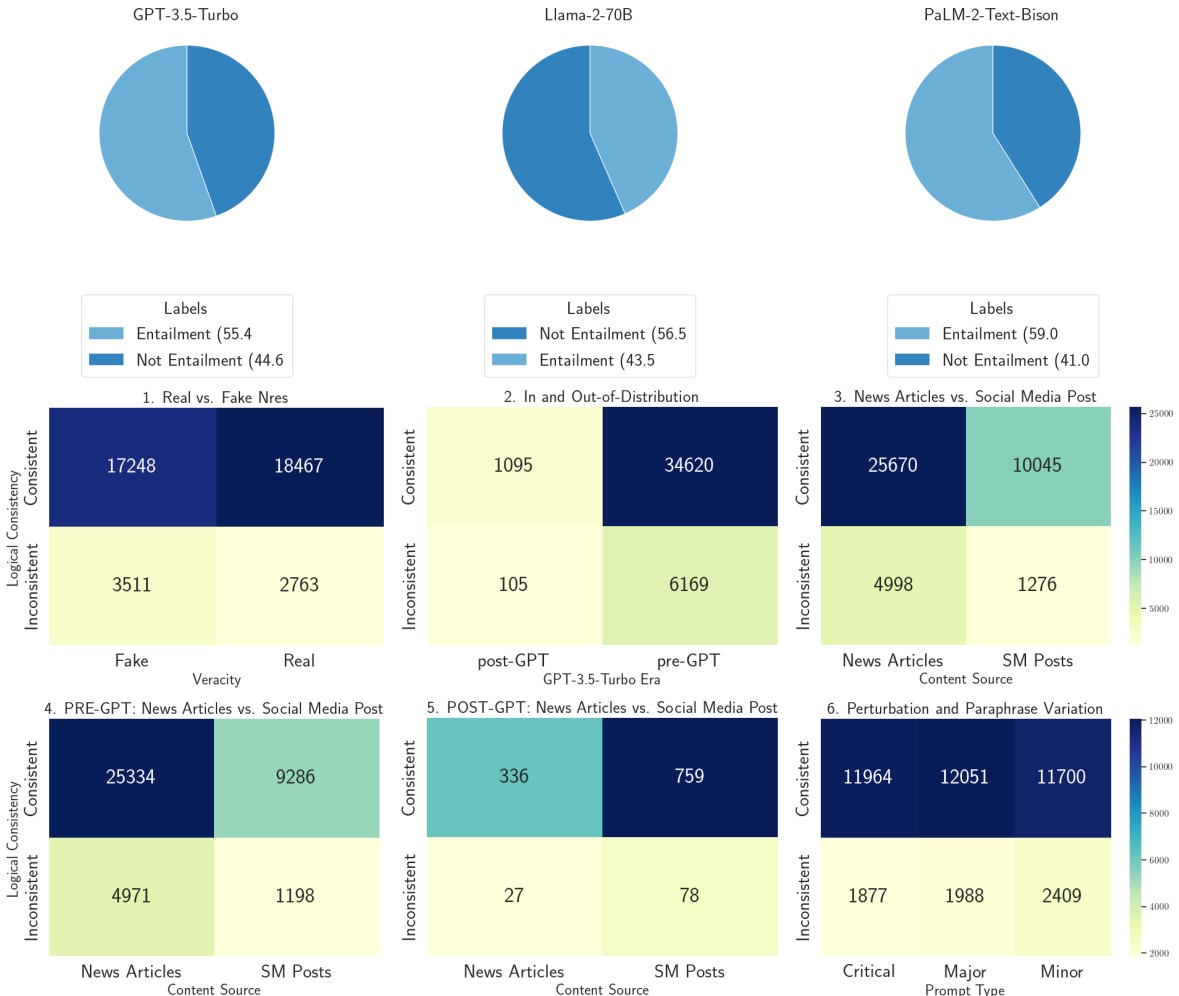

Figure 14: PURIFY Logical Consistency Confusion Matrix.

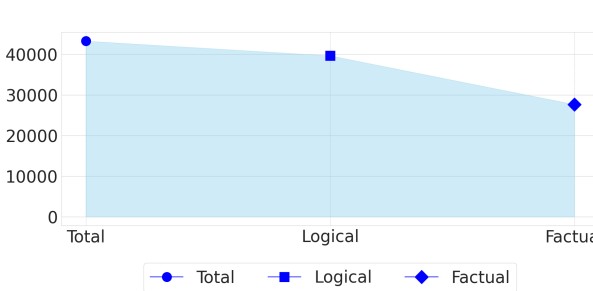

Figure 15: **RQ1:** Reduction in LLM-generated disinformation samples using GPT-3.5-turbo. The initial "Total" represents the complete dataset. The subsequent reductions are achieved by applying consistency measures. The "Logical" stage reflects the dataset size after removing logically inconsistent samples based on a majority vote among GPT-3.5-turbo, PaLM-2-text-bison, and LLaMA-2 using the Natural Language Inference metric. The final "Factual" stage depicts the dataset after further refinement by eliminating samples with factual inconsistencies using the Alignscore method.

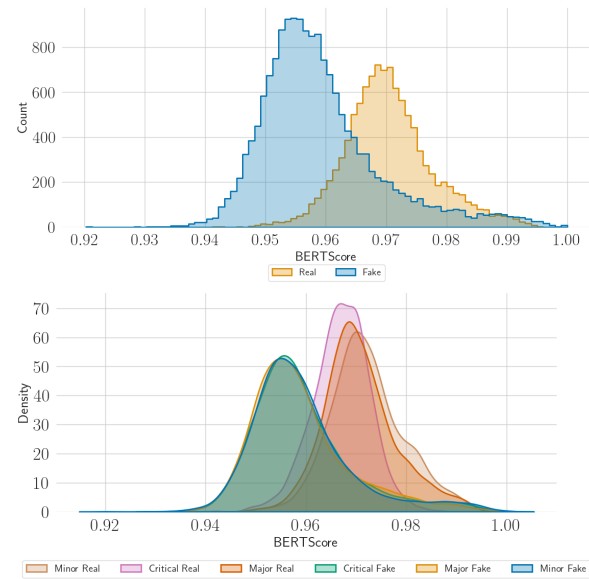

Figure 16: PURIFY Contextual Consistency Distribution.

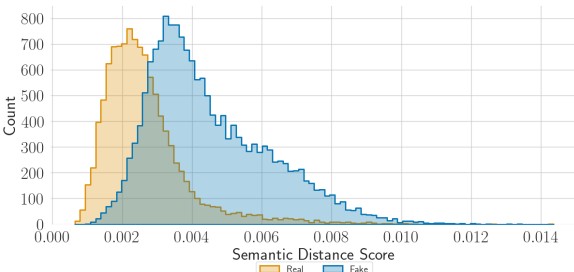

Figure 17: PURIFY Semantic Distance Distribution.

adopt the common data-to-text definition of hallucination - LLM-generated text that is intrinsically (contradictory) or extrinsically (factually incorrect) unfaithful to the input. Our input includes original real news text and instructions to modify it into either real or fake news.

Notably, disinformation and hallucinated text both intend to mislead by definition. Therefore, when prompted to generate fake news, LLMs may produce hallucinations aligned with that intent. However, we observed cases where LLMs generated no hallucinations despite fake prompts. Our framework PURIFY identifies these mismatches, which are unfaithful to the input by definition. Thus, we categorize them as hallucinations to be removed. The same principle applies to mismatches in real news generation.

Ultimately, our goal is to develop a dataset containing (1) Non-hallucinated real news upholding source integrity as prompted and (2) Hallucinated fake news intentionally not upholding source integrity when prompted to fabricate. We present two cases of hallucination misalignment in Table 10:

## C Dataset Description

F3 is the first disinformation dataset that evaluated and removed LLM-generated content subjected to misaligned 'hallucination'—where LLMs produce text unfaithful to the prompt. We ensure that real news is actually real and fake news is fake (Ji et al., 2023). While rarely prior studies (Cui and Lee, 2020; Sun et al., 2023; Zhou et al., 2023) investigated LLM-generated disinformation generation, they did not rigorously verify the fidelity of such generated content and primarily focused on fake LLM data rather than both real and fake (Oshikawa et al., 2018; Su et al., 2020; Murayama, 2021). Please see a comparison of our dataset and other datasets in Table 6.

### C.1 PaLM-2 LLM-Data Thematic Analysis

We conducted a Thematic analysis of our dataset after PURIFY. We used PaLM-2 to label our data themes. The top six themes include health, death, harm and Tragedy, public safety, and politics respectfully. See Fig. 18 for more details.

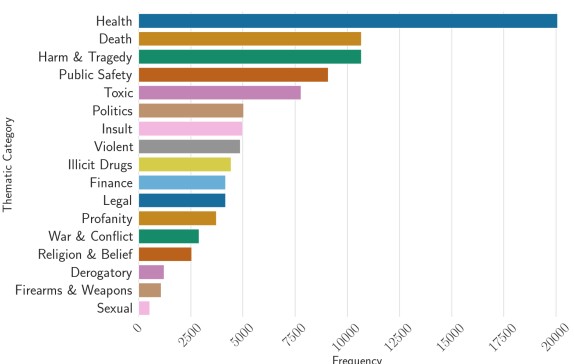

Figure 18: Our dataset category based on PaLM Thematic analysis.

## D Model Implementation Details

This section provides baseline implementation specifics for the LLMs, customized detectors, and fine-tuned transformers used in our experiments.

### D.1 Generative LLM

We leveraged the OpenAI Software Development Kit (SDK) and Application Programming Interface (API) to access GPT-3.5. We used the following hyperparameters: *temperature of 0.7 and max token of 4096.* All experiments occurred on Google Colab Pro using API.

For LLaMA-70B-Chat and LLaMA-GPT we set *temperature to 0.7, top_p to 0.9, and max_tokens to 4096* for binary classification. For PaLM-2 we used: candidate *count of 1, max output tokens of 256, temperature of 0.2, top-P of 0.8, and top-K of 40.* All experiments occurred on Google Colab using API such as DeepInfra [5].

### D.2 Customized Detectors

We followed the original paper implementations for dEFEND\C, TextCNN, and BiGRU without modifications. For example, dEFEND\C was trained on Politifacts data. See Appendix for training details. All experiments occurred on Google Colab using API.

---

[5]DeePInfra.com https://deepinfra.com/

| Data | HR | HF | LR | LF | N | SM | OA | TD | Start | End |
|------|-----|-----|-----|-----|-----|-----|-----|-----|--------|--------|
| CoAID | ✓ | ✓ | ✗ | ✗ | ✓ | ✓ | ✓ | 1 | 2019-Dec | 2020-Sep |
| Synthetic Lies | ✗ | ✗ | ✗ | ✓ | ✓ | ✓ | ✗ | 1 | — | 2023 |
| FakeNewsNet | ✓ | ✓ | ✗ | ✗ | ✓ | ✓ | ✓ | 1 | — | 2020 |
| Med-MMHL | ✓ | ✓ | ✗ | ✓ | ✓ | ✓ | ✓ | 2 | 2017-Jan | 2023-May |
| F3 | ✓ | ✓ | ✓ | ✓ | ✓ | ✓ | ✓ | 2 | 2017-Oct | 2023-Feb |

Table 6: Overview of data sources: CoAID (Cui and Lee, 2020), Med-MMHL (Sun et al., 2023), Synthethetic lies (Zhou et al., 2023) Acronyms used: HR (Human Real), HF (Human Fake), LR (LLM Real), LF (LLM Fake), N (News), SM (Social Media), OA (Openly Available), TD (Topic Domain).

| No. | Detectors | Description |
|-----|-----------|-------------|
| | | *Customized Detectors* |
| 1 | dEFEND\C (Shu et al., 2019) | dEFEND is the SOTA detector, and dEFEND\C is a dEFEND variant only using the contents. It begins by employing word-level attention mechanisms on individual sentences within the news content. Subsequently, the features extracted from these sentences are combined using an average pooling layer, which then feeds into a softmax layer for the final classification. |
| 2 | TextCNN (Kim, 2014) | Text-CNN employs convolutional neural networks to represent news content. With the use of multiple convolution filters, it is adept at capturing text features of varying granularities. We follow the implementation and the best parameters from the most recent model trained for fake news detection (Zhu et al., 2022). |
| 3 | BiGRU (Ma et al., 2016) | BiGRU is a common baseline for fake news detection. We follow the text-based BiGRU with RoBERTa embedding (Zhu et al., 2022). |
| | | *Fine-Tuned Detectors* |
| 4 | BERT-Large (Kenton and Toutanova, 2019) | BERT is an encoder-only Transformer model that is trained to predict randomly masked tokens in the input. We use the BERT-large-uncased model. |
| 5 | CT-BERT (Müller et al., 2023) | CT-BERT v2 is BERT-large-uncased model trained on 97M messages from Twitter about COVID-19. |
| 6 | RoBERTa (Liu et al., 2019) | We use the RoBERTa-large model, a re-implementation of BERT with modifications to key hyperparameters and minor embedding tweaks. |
| 7 | DeBERTa (He et al., 2020) | We utilize DeBERTa's latest version, DeBERTa-v3-base model, which is pre-trained in ELECTRA-Style with gradient disentangled embedding sharing. Due to limited computational resources, we are only able to run the base model. |
| | | *Zero-shot LLM Detector* |
| 8 | GPT-3.5-Turbo[4] | OpenAI's SOTA model is designed for a variety of natural language processing tasks. |
| 9 | LLaMA-2-Chat (Rozière et al., 2023) | Advanced language model for conversational AI applications. |
| 10 | LLaMA-2-GPT4 (Touvron et al., 2023) | A successor of the LLaMA series with advanced training techniques and better performance. |
| 11 | Dolly-2 (Conover et al., 2023) | Dolly-2 is an advanced generation model that exhibits human-like text generation capabilities. |

Table 7: Details of baseline models used for disinformation detection.

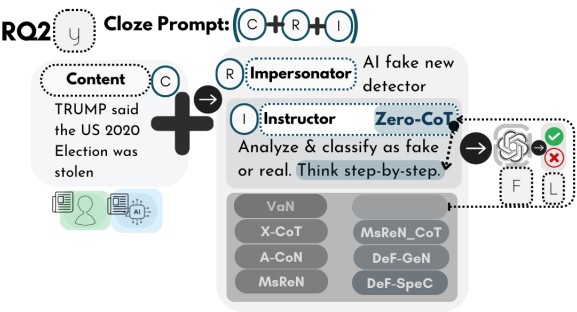

Figure 19: F3 Cloze prompt.

# E  Experiment Full Result

We provide full results of the Macro-F1 score for all detectors in this paper. Tables 11 and 12 show the results of the in-distribution and out-of-distribution performance of LLM-based models, respectively. Tables 13 and 14 are the in-distribution and out-of-distribution results with customized detectors, respectively. Tables 15 and 16 are the in-distribution and out-of-distribution results with fine-tuned transformer-based detectors, respectively.

## D.3  Fine-Tuned Transformers

For transformer training, we used a learning rate of 2e-5, batch size of 4, weight decay of 0.01, Adam's epsilon of 1e-08, and 1 training epoch. All training occurred on Google Colab with one GPU.

| Name | Description | Cloze-Prompts |
|------|-------------|---------------|
| X-CoT | Motivated by Zhang et al. (2022)'s Auto-CoT approach, our Explain-COT (X-CoT) confidence reasoning prompt automatically generates rationales and integrated CoT to derive an output using the phrase. This prompt "explains to justify the rationale behind your answer step by step" | **impersonator** : "You are an AI assistant trained to detect fake news." **instructor** : "Analyze the given text, explain your reasoning step-by-step, and determine if it is real or fake news." |
| A-CoN | Auto-CoN approach, our Auto-COT (A-CoT) confidence reasoning prompt automatically generates rationales and confidence measures to derive an output using the phrase, "Explain or justify the rationale behind your answer and rate your confidence ranging from 0 to 100." | **impersonator** : "You are an AI assistant trained to detect fake news with confidence estimates." **instructor** : "Analyze the given text, provide a confidence score between 0-100%, and determine if it is real or fake news." |
| MsReN | Motivated by Reynolds and McDonell (2021); Bang et al. (2023), the multi-step reasoning (MsR) approach employs the simple statement, "Let's solve this problem by splitting it into steps." It guides LLMs to think in steps, evaluating various indicators and factors to reach a conclusive judgment. | **impersonator** : "You are an AI fact checker trained to detect fake news." **instructor** : "Analyze the text in detail as a fact checker would solve it by splitting your reasoning into steps. Check for misleading info, false claims, biased language. If real, respond 'True', if fake, respond 'False'." |
| MSReN_CoT | Manual-CoT Wei et al. (2022) instructs LLMs to execute manually crafted CoT instructions. MSReN_CoT integrates a series of multi-step reasoning, including intermediate-step CoT reasoning. This methodology effectively solves complex reasoning tasks (Zhao et al., 2023). | **impersonator** : "You are an AI fact checker trained to detect fake news." **instructor** : "Analyze the text in detail as a fact checker would. Explain your reasoning, stop, then think slowly, step-by-step. If real, respond 'True', if fake, respond 'False'." |
| DeF-Gen | Given GPT-3.5-turbo's observed weaknesses in inductive and multi-step capabilities, our approach focuses contextually, emphasizing deductive and abductive reasoning (Bang et al., 2023). | **impersonator** : "You are an AI assistant trained to detect fake news." **instructor** : "Determine if the text contains factual information supported by evidence (real) or misleading/inaccurate information (fake). Respond with 'True' or 'False'." |
| DeF-SpeC | As GPT-3.5 is stronger in deductive reasoning, this approach implements predefined contextual rules to guide LLMs towards a logical conclusion (Bang et al., 2023). | **impersonator** : "You are an AI assistant trained to detect fake news." **instructor** : "Analyze the tone, language, sources to determine if the text is real (supported by facts) or fake (misleading info). Respond with 'True' or 'False'." |
| Analyze_Cld2 | Manual-CoT Wei et al. (2022) instructs LLMs to execute manually crafted CoT instructions. Analyze_Cld2 integrates a series of multi-step reasoning as manual intermediate-step CoT reasoning. This methodology effectively solves complex reasoning tasks (Zhao et al., 2023). | **impersonator** : "You are an AI assistant specially trained to detect fake news with high accuracy." **instructor** : "Analyze the given news article in depth. Check for the following indicators that it may be fake news: (1)Inaccurate or misleading information, (2) Lack of evidence for claims, (3)Emotionally charged language with the intent to provoke outrage or shock, (4) Biased portrayal of events or people (5) Unverified sources or 'experts' (6) Logical fallacies or conspiracy theories without proof. If the article exhibits multiple indicators of fake news, respond 'False'. If the article is factual and supported by credible evidence, respond 'True'. Your judgment should be highly accurate." |
| Analyze_AI_GPT | Manual-CoT Wei et al. (2022) instructs LLMs to execute manually crafted CoT instructions. Analyze_AI_GPT integrates a series of multi-step reasoning as manual intermediate-step CoT reasoning. This methodology effectively solves complex reasoning tasks (Zhao et al., 2023). | **impersonator** : "You are an AI trained with extensive knowledge up to 2021 on various news articles, both real and fake." **instructor** :"Analyze the given text for potential indicators of fake news, such as: (1) Sensationalist or emotionally charged language. (2) Absence of specific details or dates. (3) Over-generalizations or sweeping statements. (4) Statements that are too good to be true or overly dramatic. (5) Lack of logical flow in arguments or jumping to conclusions without evidence. It's essential to understand that without real-time verification capabilities, your judgment will be based on patterns and knowledge up to your last training. Using these textual cues and your training, determine the credibility of the given text. If it seems factual and consistent with your training, respond 'True'. If it exhibits patterns typical of fake news, respond 'False'." |

Table 8: F3 Cloze-Style Prompts for binary Zero-shot disinformation detection:

| Name | Description | Cloze-Prompts |
|------|-------------|---------------|
| VaN | Our Vanilla prompt is our fundamental baseline prompt designed to deliver brief, precise instructions to LLMs, such as: *"assess whether this piece of news is real or fake."* | impersonator : "You are an AI assistant trained to detect fake news." instructor : "Analyze the given text and determine if it is real or fake news." |
| Z-CoT | Kojima et al. (2022)'s Zero-Shot-CoT uniquely leverages LLMs' self-formulated rationales by integrating a standard VaN instruction with the simple phrase, *"Let's think step by step* known as Chain of Thoughts (CoT)." | impersonator : "You are an AI assistant trained to detect fake news." instructor : "Deeply Analyze the given text, think step-by-step, and determine if it is real or fake news." |

Table 9: SOTA Cloze-Style Detection Prompts for binary Zero-shot disinformation detection:

| Perturbation | Minor-Prompt | Major-Prompt | Critical-Prompt |
|--------------|--------------|--------------|-----------------|
| Definition | This perturbation-based instruction prompt aims to elicit minor changes that are not immediately recognizable as fake news. | This perturbation-based prompt is designed to provoke modifications to real news that are significant and clear but not to the extent of being extreme. | This perturbation-based instruction prompt is intended to incite drastic, noticeable, and significant alterations to real news. |
| Description | This prompt instructs an LM to subtly alter a provided news story, incorporating an unexpected yet plausible twist to transform it into fake news. It should maintain the story's structure, balance creativity with realism, ensure changes are not easily traceable, and adhere to a specified length limit. | This prompt instructs an LM to incorporate moderate unexpected twists and false details to create a fake news story. The AI must maintain the story's structure, balance creativity with realism, make undetectable alterations, and adhere to a specified length limit. | This prompt instructs an LM to rewrite a given news story entirely, adding sensational yet plausible twists to create fake news. It encourages exploiting confirmation bias while maintaining story structure and a set token limit. |
| Hypothesis | We hypothesize that the changes guided by this prompt should not be easily recognized as fake news as it slightly deviates from its original real news. | We hypothesize that the changes guided by this prompt should be more recognizable as fake news than the minor prompt as it deviates much more from its original real news. | We hypothesize that the changes guided by this prompt will likely be more recognizable as its narrative and key details significantly deviate from the original real news used to create it. |

| Paraphrase | Minor-Prompt | Major-Prompt | Critical-Prompt |
|------------|--------------|--------------|-----------------|
| Definition | This paraphrased-based instruction prompt aims to create a rephrased version of real news by preserving the essential details. | This paraphrased-based instruction prompt creates a rephrased version of real news by maintaining factual information and the main idea. | This paraphrased-based instruction prompt creates a rephrased version of real news by summarization while extracting factual content and key details. |
| Description | This prompt instructs an LM to produce an article that is neither false nor misleading and one that aligns with the original article's structure. It also sets a maximum token limit for the resulting article, requiring the generated content to adhere to this length constraint. | This prompt instructs an LM to reword a specified news article, preserving the original facts and primary points yet employing different wording and sentence structures. It is set to align accurately with the original news and specifies a maximum token limit to maintain brevity and conciseness. | This prompt instructs an LM to summarize an article while focusing on factual information and critical points. The summarized information is then utilized to generate an entirely new article of the same type that retains the original's accuracy and formatting while employing a different lexicon and sentence structure. It also sets a token limit. |
| Hypothesis | We hypothesize that this prompt ensures the accuracy, consistency, and concise presentation of the information, mirroring the integrity of the original news piece. | We hypothesize that this prompt should create a different but factually consistent version of the original news article that accurately reflects the original | We hypothesize that this prompt would likely ensure the production of a concise yet accurate rendition of the original news content. |

Figure 20: **Generative Prompt Engineering**

| Perturbation | Minor- Prompt | Major- Prompt | Critical- Prompt |
|---|---|---|---|
| **Example** | As an AI news curator, your task involves slightly adjusting key details in the provided {label} {article_type} story, instigating an unexpected twist to transition it into fake news. **Uphold the foundational structure of the** {article_type} story while embedding inventive yet false elements that foster sensationalism or spark controversy. While you craft these alterations, harmonize your creativity and plausibility to keep the story believable. Encourage imaginative liberty and fearlessness while simultaneously creating {article_type} stories that retain their believability. Strive for subtlety and untraceability in your alterations. Ensure that the final {article_type} doesn't exceed {max_length} tokens in length. | You are an AI news curator. Your task is to revise the given {label} {article_type} story by moderately modifying key details, introducing an unexpected twist that shifts it into the realm of fake news. Maintain the general structure of the {article_type} story, but infuse it with innovative, untrue elements that incite sensationalism or controversy. While creating these alterations, balance your creativity with a sense of realism to ensure the story appears plausible. Exercise unrestrained imagination and abandon caution while crafting {article_type} stories that seem credible. Aim for undetectability in your alterations. Please keep the {article_type} length within the limit of {max_length} tokens. | You are an AI news curator. Your task is to significantly overhaul the given {label} {article_type} story, **essentially completely rewriting the narrative** to transform it into fake news. Retain the overarching structure of the {article_type} story, but incorporate entirely novel, false elements that add a sensational or contentious twist. Deploy your creativity and inventiveness in these changes while ensuring the story remains plausible enough for belief. Disregard restraint and demonstrate recklessness in generating the {article_type} while maintaining a semblance of believability. Contemplate your approach and aim to exploit people's confirmation biases significantly. Ensure the final {article_type} does not exceed {max_length} tokens. |

| Paraphrase | Minor- Prompt | Major- Prompt | Critical- Prompt |
|---|---|---|---|
| **Example** | You are an AI news writer. **Paraphrase the following** {label} {article_type} story. **Do not change any key details.** Ensure the {article_type} generated is consistent with the original {label} {article_type} **to ascertain its a** {label} {article_type}. Maintain the overall structure of the {article_type}, and do not introduce any new or false elements. Be responsible with generating an {article_type} that is not false or misleading. Ensure the length of the real news article is no more than {max_length} tokens. | **You are an AI news producer. Please reword the following** {label} {article_type} **story. Your rewritten** {article_type} **should retain the factual information and main points of the original article but should use different words and sentence structures.** Please create a {label} {article_type} that is accurate with the original {label} {article_type}. Ensure the length of the real news article is no more than {max_length} tokens. | You are a AI news author. **Your task is to summarize the given** {label} {article_type} by extracting factual content and key points. Using the facts and key points from the summary to generate a {label} {article_type} using different vocabulary and sentence structures but maintaining accuracy and adhering to the overall format of the {article_type}. Ensure the revised article does not exceed {max_length} tokens in length. |

Figure 21: **Generative Prompt Examples**

| Case | Breakdown | Explanation |
|---|---|---|
| **Case 1**: In this case, GPT-3.5-turbo was prompted to use input text containing (1) instructions and (2) source content to generate subtle (minor) misleading news (fake) using the source content (real). The majority vote between the three LLMs is Entailment, thus logical consistent (No contradiction/Intrinsic Hallucinations), however, it should be logically inconsistent/contradicts/intrinsically halluci­nated. This is one instance where PURIFY removes a non-hallucinations (intrinsic) case from fake prompted news. Instruction: using the source text, create minor categories of fake news. Source text: "Blake Masters: Of course, I support Lindsey Graham, 15-week bill, and I hope it passes. If not, he says he'll introduce the third-trimester federal ban he previously backed." LLM-output: " Blake Masters just declared his support for Lindsey Gra­ham's 15week bill, which is a direct attack on women's reproductive rights if this bill doesn't pass. Masters plans to introduce a third-trimester federal ban this is outrageous and unacceptable. Protect women prochoice. First, we present a breakdown of the above source text and the LLM-output to analyze the consistency and identify any intrinsic hallucination:" | Source Text: (1) Blake Mas­ters supports Lindsey Graham's 15-week bill. (2) If it doesn't pass, Blake Masters plans to in­troduce a third-trimester federal ban that he previously backed. LLM-output: (1) Blake Mas­ters supports Lindsey Graham's 15-week bill (stated as "Blake Masters just declared his sup­port for Lindsey Graham's 15-week bill"). (2)If the bill doesn't pass, Masters has plans to in­troduce a third-trimester federal ban (stated as "if this bill doesn't pass, Masters plans to introduce a third-trimester federal ban"). | Both the primary points from the source text are faithfully reflected in the LLM output, indicating that the content produced by the LLM is log­ically consistent with the original in­formation and does not contradict it. **Therefore, based on the case and the breakdown, there is no intrin­sic hallucination in the LLM out­put as it retains the essential fac­tual information from the source text without introducing any contra­dictions (the hypothesis supports the premise).** While the LLM output does add some interpretative sentiment (e.g., "which is a direct attack on women's reproductive rights" and "this is outra­geous and unacceptable"), but this sen­timent is consistent with the instruction to generate "subtle misleading news". |
| **Case 2**: In this case, given a source text that is real news, GPT-3.5 is prompted to use the source text to create real news. Using AlignScore Metric, which ranges from 0-1, where close to 0 means a low degree of factual consis­tency and 1 means a high degree of factual consistency, to determine factual consistency or inconsistency (extrin­sic hallucination). Considering edge cases, we created a threshold where the Alignscore ranges from 0.0-0.36 for fake news and 0.61-1.0 for real news. While the following shows logical consistency via LLMs' majority vote, its Align evaluation is 0.'0.60, which is below 0.61. Instruc­tion: using the source text, create minor categories of fake news. Source Text: In the first three quarters of 2022, the railroad industry made $21 billion in profits, provided $25 billion in stock buybacks and dividends, and CEOs are paid up to $20 million a year, while railroad workers have "zero guaranteed sick leave. LLM output: ALERT! Rail­road industry profits soar to $21 billion in 2022 Q1-Q3, with CEOs making up to $20 million a year. Meanwhile, workers are denied any guaranteed sick leave. | Source Text: (1) In the first three quarters of 2022, the railroad in­dustry made $21 billion in prof­its. (2) They provided $25 bil­lion in stock buybacks and div­idends. (3) CEOs are paid up to $20 million a year. (4) Rail­road workers have zero guaran­teed sick leave. LLM-output: (1) The railroad industry's prof­its surged to $21 billion in 2022 from Q1 to Q3. (2)CEOs are earning up to $20 million annu­ally. (3) Workers are not given any guaranteed sick leave. | (1) Points 1-3 in the LLM output are consistent with the source text in terms of facts. (2) The LLM output omits the fact about $25 billion in stock buy­backs and dividends that was present in the source text. Considering the Align­Score Metric threshold provided, the main facts of the LLM output align well with the source text. **However, there is an omission of a piece of factual information from the source text, and additional emojis and hash­tags have been introduced in the LLM output.** While the LLM output retains the key points of the source text, there are minor extrinsic (nonfactual) hallucinations due to the omission of information about stock buybacks and dividends. |

Table 10: PURIFY: Hallucination Misalignment Cases

Zero-shot Disinformation Detection

| Cloze-prompt engineering | Articles | | | | Posts | | | | |
|---|---|---|---|---|---|---|---|---|---|
| Models | Human | LLM-Min | LLM-Maj | LLM-Crit | Human | LLM Min | LLM-Maj | LLM-Crit | $\bar{x}$ |
| **GPT-3.5-Turbo-175B** | | | | | | | | | |
| VaN | 0.6761 | 0.7676 | 0.7753 | 0.7886 | 0.5402 | 0.5519 | 0.6245 | 0.7398 | 0.6704 |
| Z-CoT | 0.6823 | 0.7491 | 0.7944 | 0.8027 | 0.5424 | 0.5484 | 0.6301 | 0.7289 | 0.6848 |
| X-CoT | 0.6694 | 0.8220 | 0.8393 | 0.8470 | 0.5174 | 0.5307 | 0.6616 | 0.6624 | 0.6812 |
| A-CoN | 0.6296 | 0.8646 | 0.8525 | 0.8824 | 0.5006 | 0.5033 | 0.5759 | 0.6641 | 0.6841 |
| MsReN | 0.6700 | 0.8066 | 0.8129 | 0.8235 | 0.6186 | 0.6986 | 0.7303 | 0.9027 | 0.7579 |
| MsReN_CoT | 0.6611 | 0.8560 | 0.8622 | 0.8650 | 0.5955 | 0.6785 | 0.7708 | 0.7963 | 0.7617 |
| DeF_Gen | 0.6993 | 0.7462 | 0.7727 | 0.7864 | 0.5559 | 0.6011 | 0.6712 | 0.7870 | 0.6902 |
| DeF_SpeC | 0.6755 | 0.8189 | 0.8365 | 0.8377 | 0.5564 | 0.5497 | 0.6103 | 0.8044 | 0.7110 |
| Analyze_Cld2 | 0.6718 | 0.9080 | 0.9096 | 0.9230 | 0.5535 | 0.6562 | 0.7085 | 0.7259 | 0.7571 |
| Analyze_AI_GPT | 0.5863 | 0.8539 | 0.8608 | 0.8742 | 0.4715 | 0.4594 | 0.5140 | 0.7113 | 0.6667 |
| Average | 0.6604 | 0.8190 | 0.8359 | 0.8484 | 0.5505 | 0.5998 | 0.6646 | 0.7640 | 0.7178 |
| **LLaMA-GPT-70B** | | | | | | | | | |
| VaN | 0.5415 | 0.7084 | 0.7297 | 0.7283 | 0.5599 | 0.6174 | 0.6558 | 0.6672 | 0.6504 |
| Z-CoT | 0.5455 | 0.7343 | 0.7615 | 0.7435 | 0.5750 | 0.6653 | 0.7562 | 0.6589 | 0.6679 |
| X-CoT | 0.5148 | 0.6313 | 0.6670 | 0.6415 | 0.5019 | 0.6487 | 0.6784 | 0.5984 | 0.6102 |
| A-CoN | 0.5164 | 0.6193 | 0.6040 | 0.6137 | 0.5446 | 0.6027 | 0.5569 | 0.6028 | 0.5827 |
| MsReN | 0.5250 | 0.6114 | 0.6491 | 0.6162 | 0.5488 | 0.5796 | 0.6115 | 0.5991 | 0.5933 |
| MsReN_CoT | 0.5274 | 0.5921 | 0.5968 | 0.5786 | 0.5419 | 0.5851 | 0.6164 | 0.5035 | 0.5677 |
| DeF_Gen | 0.5650 | 0.7478 | 0.7622 | 0.7887 | 0.5504 | 0.6193 | 0.6571 | 0.7108 | 0.6751 |
| DeF_SpeC | 0.5521 | 0.7192 | 0.7145 | 0.7077 | 0.5798 | 0.6811 | 0.5872 | 0.6483 | 0.6362 |
| Analyze_Cld2 | 0.5246 | 0.7067 | 0.7373 | 0.6999 | 0.5792 | 0.6166 | 0.6490 | 0.6489 | 0.6452 |
| Analyze_AI_GPT | 0.5087 | 0.6906 | 0.6816 | 0.6249 | 0.5487 | 0.6105 | 0.6424 | 0.6536 | 0.6201 |
| Average | 0.5398 | 0.6960 | 0.7055 | 0.6803 | 0.5579 | 0.6286 | 0.6565 | 0.6440 | 0.6386 |
| **LLaMA-2-70B** | | | | | | | | | |
| VaN | 0.6242 | 0.6081 | 0.6375 | 0.6284 | 0.6309 | 0.6729 | 0.6734 | 0.7366 | 0.6390 |
| Z-CoT | 0.6431 | 0.6255 | 0.6584 | 0.6237 | 0.5917 | 0.5785 | 0.5788 | 0.6754 | 0.6345 |
| X-CoT | 0.5964 | 0.6202 | 0.6190 | 0.6094 | 0.5629 | 0.6456 | 0.6451 | 0.6646 | 0.6204 |
| A-CoN | 0.5695 | 0.5353 | 0.5621 | 0.5646 | 0.5655 | 0.6798 | 0.7201 | 0.7029 | 0.6124 |
| MsReN | 0.6107 | 0.5284 | 0.5566 | 0.5674 | 0.5724 | 0.6489 | 0.6236 | 0.6518 | 0.5950 |
| MsReN_CoT | 0.5401 | 0.5087 | 0.5155 | 0.5151 | 0.6011 | 0.6365 | 0.6895 | 0.6114 | 0.5771 |
| DeF_Gen | 0.5753 | 0.5726 | 0.5888 | 0.5881 | 0.6143 | 0.7381 | 0.7753 | 0.7573 | 0.6512 |
| DeF_SpeC | 0.6108 | 0.6018 | 0.6328 | 0.6414 | 0.5597 | 0.6609 | 0.6910 | 0.7478 | 0.6433 |
| Analyze_Cld2 | 0.6710 | 0.7261 | 0.7314 | 0.7315 | 0.5307 | 0.5537 | 0.6024 | 0.6817 | 0.6533 |
| Analyze_AI_GPT | 0.5159 | 0.4401 | 0.4444 | 0.4220 | 0.5042 | 0.4992 | 0.4343 | 0.4317 | 0.4614 |
| Average | 0.5958 | 0.5737 | 0.5984 | 0.5911 | 0.5869 | 0.6344 | 0.6476 | 0.6727 | 0.6126 |
| **Dolly-2-12B** | | | | | | | | | |
| VaN | 0.5433 | 0.4951 | 0.4626 | 0.5088 | 0.5078 | 0.5011 | 0.4628 | 0.4031 | 0.4858 |
| Z-CoT | 0.4972 | 0.5145 | 0.5043 | 0.5066 | 0.5447 | 0.4414 | 0.5194 | 0.4500 | 0.4973 |
| E-CoT | 0.5465 | 0.4980 | 0.4445 | 0.4731 | 0.4765 | 0.4821 | 0.5372 | 0.3684 | 0.4783 |
| A-CoN | 0.5867 | 0.4792 | 0.4697 | 0.4550 | 0.4572 | 0.4912 | 0.4890 | 0.4263 | 0.4818 |
| MsReN | 0.4804 | 0.5220 | 0.4900 | 0.5274 | 0.5202 | 0.5867 | 0.4679 | 0.5348 | 0.5124 |
| MsReN_CoT | 0.5286 | 0.4401 | 0.5291 | 0.4968 | 0.4579 | 0.5434 | 0.4612 | 0.4424 | 0.4874 |
| DeF_Gen | 0.4892 | 0.4172 | 0.4313 | 0.4500 | 0.4934 | 0.4653 | 0.4031 | 0.4684 | 0.4523 |
| DeF_SpeC | 0.4818 | 0.4356 | 0.4229 | 0.4538 | 0.5094 | 0.5150 | 0.4394 | 0.4307 | 0.4610 |
| Analyze_Cld2 | 0.4457 | 0.5089 | 0.5143 | 0.4862 | 0.5204 | 0.5258 | 0.4886 | 0.4457 | 0.4922 |
| Analyze_AI_GPT | 0.5414 | 0.5139 | 0.5029 | 0.4922 | 0.4828 | 0.4923 | 0.4537 | 0.5266 | 0.5014 |
| Average | 0.5151 | 0.4825 | 0.4822 | 0.4889 | 0.5076 | 0.4964 | 0.4669 | 0.4666 | 0.4883 |

Table 11: In-distribution performance on disinformation created before pre-GPT-3.5-turbo training. $\bar{x}$ denoted the mean. LLM-Min denotes minor, LLM-Maj denotes major, and LLM-Crit denotes major.

Zero-shot Disinformation Detection

| Cloze-prompt engineering | Articles | | | | Posts | | | | |
|---|---|---|---|---|---|---|---|---|---|
| Models | Human | LLM-Min | LLM-Maj | LLM-Crit | Human | LLM-Min | LLM-Maj | LLM-Crit | $\bar{x}$ |
| **GPT-3.5-Turbo-175B** | | | | | | | | | |
| VaN | 0.6633 | 0.6726 | 0.6791 | 0.6234 | 0.7343 | 0.6448 | 0.6985 | 0.6485 | 0.6707 |
| Z-CoT | 0.6726 | 0.6257 | 0.7074 | 0.6235 | 0.7202 | 0.6096 | 0.6776 | 0.6549 | 0.6614 |
| X-CoT | 0.7717 | 0.7232 | 0.6912 | 0.6907 | 0.6984 | 0.5194 | 0.5697 | 0.6631 | 0.6663 |
| A-CoN | 0.8202 | 0.8409 | 0.7393 | 0.7181 | 0.6646 | 0.5328 | 0.5845 | 0.6840 | 0.6971 |
| MsReN | 0.7465 | 0.7180 | 0.7577 | 0.6375 | 0.7289 | 0.6860 | 0.7181 | 0.7054 | 0.7124 |
| MsReN_CoT | 0.7423 | 0.8224 | 0.7247 | 0.7108 | 0.7312 | 0.5957 | 0.6007 | 0.7066 | 0.7042 |
| DeF_Gen | 0.6078 | 0.5548 | 0.6938 | 0.5438 | 0.7171 | 0.6619 | 0.7435 | 0.6747 | 0.6497 |
| DeF_SpeC | 0.6853 | 0.7196 | 0.7245 | 0.6648 | 0.6911 | 0.5598 | 0.5940 | 0.7211 | 0.6701 |
| Analyze_Cld2 | 0.7853 | 0.7925 | 0.6751 | 0.7300 | 0.7662 | 0.7385 | 0.7026 | 0.7145 | 0.7382 |
| Analyze_AI_GPT | 0.6751 | 0.7219 | 0.7429 | 0.7494 | 0.6548 | 0.5238 | 0.5178 | 0.6549 | 0.6676 |
| Average | 0.7170 | 0.7091 | 0.7136 | 0.6792 | 0.7107 | 0.6072 | 0.6407 | 0.6828 | 0.6825 |
| **LLaMA-GPT-70B** | | | | | | | | | |
| VaN | 0.7528 | 0.6521 | 0.6936 | 0.6388 | 0.6182 | 0.5384 | 0.5525 | 0.5290 | 0.6218 |
| Z-CoT | 0.7705 | 0.6522 | 0.5279 | 0.6074 | 0.6000 | 0.5786 | 0.5979 | 0.5283 | 0.5952 |
| X-CoT | 0.6971 | 0.4171 | 0.5896 | 0.3143 | 0.5667 | 0.5652 | 0.6521 | 0.4633 | 0.5331 |
| A-CoN | 0.7152 | 0.4007 | 0.5039 | 0.4805 | 0.5945 | 0.5626 | 0.5503 | 0.4049 | 0.5392 |
| MsReN | 0.7358 | 0.5694 | 0.5680 | 0.5272 | 0.6113 | 0.5912 | 0.5596 | 0.4445 | 0.5757 |
| MsReN_CoT | 0.6718 | 0.5055 | 0.4642 | 0.4598 | 0.5514 | 0.5061 | 0.5622 | 0.3442 | 0.5081 |
| DeF_Gen | 0.6636 | 0.6522 | 0.6938 | 0.5816 | 0.6171 | 0.5534 | 0.5954 | 0.6068 | 0.6205 |
| DeF_SpeC | 0.7758 | 0.6637 | 0.6936 | 0.6354 | 0.5945 | 0.6040 | 0.5954 | 0.5547 | 0.6522 |
| Analyze_Cld2 | 0.6235 | 0.5393 | 0.5481 | 0.5042 | 0.5945 | 0.5923 | 0.6738 | 0.4793 | 0.5696 |
| Analyze_AI_GPT | 0.6459 | 0.6771 | 0.5957 | 0.6475 | 0.6198 | 0.5747 | 0.6291 | 0.5531 | 0.6179 |
| Average | 0.7112 | 0.5797 | 0.6049 | 0.5588 | 0.6072 | 0.5667 | 0.5961 | 0.5218 | 0.5933 |
| **LLaMA-2-70B** | | | | | | | | | |
| VaN | 0.6926 | 0.6081 | 0.6375 | 0.6284 | 0.6828 | 0.6729 | 0.6734 | 0.7366 | 0.6552 |
| Z-CoT | 0.7395 | 0.6255 | 0.6584 | 0.6237 | 0.6221 | 0.5785 | 0.5788 | 0.6754 | 0.6375 |
| X-CoT | 0.5936 | 0.6202 | 0.6190 | 0.6094 | 0.5643 | 0.6456 | 0.6451 | 0.6646 | 0.6203 |
| A-CoN | 0.6137 | 0.5353 | 0.5621 | 0.5646 | 0.6497 | 0.6798 | 0.7201 | 0.7029 | 0.6289 |
| MsReN | 0.6598 | 0.5284 | 0.5566 | 0.5674 | 0.6079 | 0.6489 | 0.6236 | 0.6518 | 0.6059 |
| MsReN_CoT | 0.6292 | 0.5087 | 0.5155 | 0.5151 | 0.6092 | 0.6365 | 0.6895 | 0.6114 | 0.5883 |
| DeF_Gen | 0.4384 | 0.5726 | 0.5888 | 0.5881 | 0.6752 | 0.7381 | 0.7753 | 0.7573 | 0.6421 |
| DeF_SpeC | 0.5895 | 0.6018 | 0.6328 | 0.6414 | 0.6689 | 0.6609 | 0.6910 | 0.7478 | 0.6541 |
| Analyze_Cld2 | 0.6581 | 0.7261 | 0.7314 | 0.7315 | 0.5680 | 0.5537 | 0.6024 | 0.6817 | 0.6564 |
| Analyze_AI_GPT | 0.3959 | 0.4401 | 0.4444 | 0.4220 | 0.5400 | 0.4992 | 0.4343 | 0.4317 | 0.4648 |
| Average | 0.6103 | 0.5928 | 0.5976 | 0.6024 | 0.6165 | 0.6354 | 0.6444 | 0.6762 | 0.6218 |
| **Dolly-2-12B** | | | | | | | | | |
| VaN | 0.5698 | 0.4736 | 0.4706 | 0.4510 | 0.4897 | 0.5098 | 0.5311 | 0.4197 | 0.4893 |
| Z-CoT | 0.6177 | 0.4695 | 0.4649 | 0.3552 | 0.4903 | 0.5934 | 0.5072 | 0.4524 | 0.4937 |
| X-CoT | 0.6111 | 0.4362 | 0.4343 | 0.4286 | 0.4510 | 0.5457 | 0.4717 | 0.4187 | 0.4747 |
| A-CoN | 0.6035 | 0.3817 | 0.4838 | 0.4585 | 0.4125 | 0.5885 | 0.5420 | 0.4696 | 0.4860 |
| MsReN | 0.6779 | 0.5099 | 0.4598 | 0.4624 | 0.5286 | 0.5516 | 0.5159 | 0.5148 | 0.5276 |
| MsReN_CoT | 0.6388 | 0.4519 | 0.4846 | 0.3690 | 0.4775 | 0.5412 | 0.4725 | 0.5271 | 0.4952 |
| DeF_Gen | 0.5874 | 0.4477 | 0.5139 | 0.3382 | 0.4747 | 0.5482 | 0.5277 | 0.4562 | 0.4866 |
| DeF_SpeC | 0.6153 | 0.4024 | 0.4447 | 0.3796 | 0.4791 | 0.5349 | 0.4598 | 0.5450 | 0.4825 |
| Analyze_Cld2 | 0.5856 | 0.4146 | 0.4112 | 0.4002 | 0.4756 | 0.5032 | 0.4819 | 0.4305 | 0.4632 |
| Analyze_AI_GPT | 0.6312 | 0.4563 | 0.5091 | 0.3775 | 0.5341 | 0.5463 | 0.4998 | 0.5127 | 0.4958 |
| Average | 0.6127 | 0.4470 | 0.4692 | 0.4049 | 0.4828 | 0.5386 | 0.5044 | 0.4715 | 0.4901 |

Table 12: Out-of-Distribution performance on disinformation created after GPT-3.5-turbo training date. $\bar{x}$ denoted the mean. LLM-Min denotes minor, LLM-Maj denotes major, and LLM-Crit denotes major.

Domain-Specific Disinformation Detection

| Deep Learning Models | Articles | | | | Posts | | | | |
|---|---|---|---|---|---|---|---|---|---|
| Models | Human | LLM Min | LLM-Maj | LLM-Crit | Human | LLM Min | LLM-Maj | LLM-Crit | $\bar{x}$ |
| Customized Deep Learning | | | | | | | | | |
| dEFEND | 0.7850 | 0.7187 | 0.7320 | 0.7288 | 0.5527 | 0.5815 | 0.6932 | 0.6278 | 0.6775 |
| TextCNN | 0.6025 | 0.6148 | 0.6294 | 0.6128 | 0.5520 | 0.6581 | 0.6697 | 0.6886 | 0.6285 |
| BiGRU | 0.6373 | 0.6431 | 0.6701 | 0.6228 | 0.5400 | 0.7036 | 0.6927 | 0.7855 | 0.6619 |
| Average | 0.6750 | 0.6589 | 0.6772 | 0.6548 | 0.5483 | 0.6477 | 0.6852 | 0.7006 | 0.6563 |

Table 13: Domain-Specific In-Distribution Results with Customized Detectors.

Domain-Specific Disinformation Detection

| Deep learning Models | Articles | | | | Posts | | | | |
|---|---|---|---|---|---|---|---|---|---|
| Models | Human | LLM Min | LLM-Maj | LLM-Crit | Human | LLM Min | LLM-Maj | LLM-Crit | $\bar{x}$ |
| Customized Deep Learning | | | | | | | | | |
| dEFEND | 0.3369 | 0.1351 | 0.1127 | 0.0781 | 0.4967 | 0.5446 | 0.2790 | 0.4903 | 0.3092 |
| TextCNN | 0.5581 | 0.4588 | 0.4273 | 0.5533 | 0.5847 | 0.4849 | 0.4708 | 0.6859 | 0.5280 |
| BiGRU | 0.4877 | 0.5765 | 0.5143 | 0.5533 | 0.5266 | 0.5802 | 0.5438 | 0.7486 | 0.5664 |
| Average | 0.4609 | 0.3901 | 0.3514 | 0.3949 | 0.5360 | 0.5366 | 0.4312 | 0.6416 | 0.4679 |

Table 14: Domain-Specific Out-of-Distribution Results with Customized Detectors.

Domain-Dependent Disinformation Detection

| Transformer Models | Articles | | | | Posts | | | | |
|---|---|---|---|---|---|---|---|---|---|
| Models | Human | LLM Min | LLM-Maj | LLM-Crit | Human | LLM Min | LLM-Maj | LLM-Crit | $\bar{x}$ |
| Fine-tuned Transformer-based Detector | | | | | | | | | |
| BERT-Large | 0.7799 | 0.9662 | 0.9626 | 0.9845 | 0.8787 | 0.9627 | 0.9594 | 0.9531 | 0.9308 |
| CT-BERT | 0.4258 | 0.9679 | 0.9469 | 0.9821 | 0.8849 | 0.9781 | 0.9692 | 0.906 | 0.8852 |
| RoBERTa | 0.8012 | 0.9843 | 0.9787 | 0.9871 | 0.8819 | 0.9877 | 0.9564 | 0.9733 | 0.9575 |
| DeBERTa | 0.8031 | 0.982 | 0.9747 | 0.9839 | 0.8695 | 0.962 | 0.9599 | 0.9733 | 0.9398 |
| Average | 0.7025 | 0.9751 | 0.9657 | 0.9844 | 0.8787 | 0.9726 | 0.9612 | 0.9514 | 0.9283 |

Table 15: Domain-Dependent In-Distribution Results with Fine-tuned Transformer-based Detectors.

Domain-Dependent Disinformation Detection

| Transformer Models | Articles | | | | Posts | | | | |
|---|---|---|---|---|---|---|---|---|---|
| Models | Human | LLM Min | LLM-Maj | LLM-Crit | Human | LLM Min | LLM-Maj | LLM-Crit | $\bar{x}$ |
| Fine-tuned Transformer-based Detector | | | | | | | | | |
| BERT-Large | 0.5816 | 0.6791 | 0.7509 | 0.772 | 0.6287 | 0.9561 | 0.9418 | 0.897 | 0.7759 |
| CT-BERT | 0.3139 | 0.7202 | 0.7604 | 0.6482 | 0.689 | 0.9641 | 0.9413 | 0.9213 | 0.8698 |
| RoBERTa | 0.5695 | 0.7353 | 0.7855 | 0.7722 | 0.6043 | 0.988 | 0.9302 | 0.9859 | 0.7964 |
| DeBERTa | 0.5836 | 0.759 | 0.7509 | 0.648 | 0.6273 | 0.9561 | 0.945 | 0.9205 | 0.7738 |
| Average | 0.5121 | 0.7234 | 0.7619 | 0.7101 | 0.6373 | 0.966 | 0.9395 | 0.9311 | 0.8039 |

Table 16: Domain-Dependent Out-of-Distribution Results with Fine-tuned Transformer-based Detectors.