# OpenReview forum: "Fighting Fire with Fire: The Dual Role of LLMs in Crafting and Detecting Elusive Disinformation"
_EMNLP/2023/Conference — EMNLP 2023 Main_

### Official Review · Reviewer_xt8f · 2023-08-03

**Soundness:** 2

**Excitement:**

2: Mediocre: This paper makes marginal contributions (vs non-contemporaneous work), so I would rather not see it in the conference.

**Missing References:**

The Science of Detecting LLM-Generated Texts by Tang et. al 2023

Can AI-Generated Text be Reliably Detected? by Sadasivan et. al 2023

Distinguishing Fact from Fiction: A Benchmark Dataset for Identifying Machine-Generated Scientific Papers in the LLM Era. by Mosca et. al 2023

**Paper Topic And Main Contributions:**

This paper proposes to check whether models can detect misinformation generated by language models and humans, by using language models to do so. They propose a suite of different prompts and datasets and show a plethora of results that indicate that in their experimental setting fine-tuned models perform better but drop in out of distribution settings.

Overall I think this paper has a strong motivation, but the experimental setting and paper presentation needs significant revision

**Questions For The Authors:**

Lines 344-356: what are the “disagreed generated texts”? Ones where they both don’t pass NLI verification? If you filter by NLI verification it’s going to be a very easy dataset, so I’m confused,. Further, what does “we confirmed that … they logically entail” mean? You manually checked 40k instances or had an NLI system do it? Did you filter all datasets or just the new FIKE dataset?

**Reasons To Accept:**

- The topic that this paper focuses on is an interesting and important topic to study. I think the core idea is interesting and useful: can LLMs do better at identifying LLM-created misinformation?
- The paper is very full of plots and figures, which although have some quirks (see the presentation issues) are generally positive and helpful
- Lots of experiments are run and reported, with a very detailed appendix.

**Reasons To Reject:**

1. My biggest concern is with the experimental design. It is unclear (see questions) and seems to be that the data was filtered in NLI style by the same models that would then assess whether it was real or fake. Further, on most datasets some models perform at near 100%, which is usually a red flag in experimental design or indicative of some bug.
2. Another major issue is that there is no outside evidence to use when assessing misinformation. For example, using the example in Figure 4: how is ChatGPT supposed to know that SF had 2x instead of 5x deaths? It seems without retrieval and outside knowledge this task is ill-formed, and is nearly meaningless - the only way to tell is to try to tell patterns in how the text is written and other heuristics (essentially similar to the watermarking style or chatGPT detection papers). I would recommend checking the line of work on retrieval based fact checking such as FEVER if the authors are not familiar if that's the angle they want, or otherwise focus on those that look at identifying LLM generated text (see references) and use some of them as baselines.
3. The paper has a significant amount of presentation flaws, such that it will require a major rewrite to be publication ready, with many updates to figures and sections.
4. Minor: FIKE-Human-AI dataset does not seem to be open source and is only “available on upon request”

**Reproducibility:**

4: Could mostly reproduce the results, but there may be some variation because of sample variance or minor variations in their interpretation of the protocol or method.

**Reviewer Confidence:**

4: Quite sure. I tried to check the important points carefully. It's unlikely, though conceivable, that I missed something that should affect my ratings.

**Typos Grammar Style And Presentation Improvements:**

Table 2 has many formatting errors, the vertical space is too compressed. Also, lots of negative vspace between captions, the text and figures are running into each other.

Similarly with many of the \paragraph tags like “Prefix Prompt Text Generation” on line 89.

Many parts of Figure 3 I cannot understand: what are MiN, MaJ, CRiT? Etc. Although they are explained later, they should be in a caption or in the figure.

What is dEFEND\C on line 229? The reference for it is several pages later

Table 3 is confusing, could you list the prompts somewhere? Currently they are descriptions of prompts, which is hard to understand and would be simpler to just write the actual prompt.

Figure 4 the critical example seems like it may be incorrect? It reads the same as the human written post (2x more drug overdoses is the same as more drug overdose deaths, and both complain about the lack of focus on drug overdoses) so it doesn’t seem like a critical difference.

Figure 5 should not be a line plot, since a change in the x-axis does not have any semantic meaning (is a categorical variable rather than continuous). I would use a bar plot or table. It’s also very difficult to understand the differences with so many lines and little space between them.

What does lines 453-455 mean? ChatGPT is worse than customized detectors but also better?

Line 456 “matching” is not usually the word to be used when something has 2x the score.

Generally sections 7.1 and 7.2 would be benefitted by more takeaways and less specific number details. I was a little overwhelmed with numbers and wasn’t sure what the main takeaway was from those sections.

---

> ### Author Rebuttal · Authors · 2023-08-29
>
> Reasons To Reject:
>
> Q.1 My biggest concern is with the experimental design. It is unclear (see questions) and seems to be that the data was filtered in NLI style by the same models that would then assess whether it was real or fake. Further, on most datasets some models perform at near 100%, which is usually a red flag in experimental design or indicative of some bug.
> (1) Reason 1.a: Thank you for raising this concern about our experimental design.
>
> - Retention Capability: GPT-3.5-turbo, when accessed via the API, does not retain prior interactions or memory, consistent with the behavior of other transformer-based LLMs. Similar to other LLMs, it is only able to retain information from the training data.
> - Model Selection Justification: We chose GPT-3.5-turbo based on its state-of-the-art zero-shot NLI performance and its expansive training data, setting it apart from previous LLMs [1, 2].
> - Distinct Tasks: The tasks of NLI assessment and detection are distinct in nature:
> (a.) NLI: Assesses the logical alignment between human-written and AI-generated text, checking if the original content is a logical subset of the generated content. Furthermore, we used 2 LLMs for this task - GPT-3.5 and Flan-T5, and took the majority vote of both to mitigate any perceived bias. Flan-T5 differs from GPT-3.5-turbo in its training and data distribution, making it suitable to contrast against GPT-3.5
> (b.) Detection: Leverages patterns inherent in the LLM's training data to classify texts as real or fake news.
>
> [1] Qin, C., Zhang, A., Zhang, Z., Chen, J., Yasunaga, M., & Yang, D. (2023). Is ChatGPT a general-purpose natural language processing task solver?. arXiv preprint arXiv:2302.06476.
>
> [2] Kojima, T., Gu, S. S., Reid, M., Matsuo, Y., & Iwasawa, Y. (2022). Large language models are zero-shot reasoners. Advances in neural information processing systems, 35, 22199-22213.
>
>
> (2) Reason.1.b Further, on most datasets some models perform at nearly 100%, which is usually a red flag in experimental design or indicative of some bug.
>
> Thank you for highlighting this observation.
> - Performance of Fine-Tuned Detectors: It's important to distinguish the performances of our models. Our customized and fine-tuned state-of-the-art (SOTA) detectors indeed achieved scores near 100%. However, such high performance, especially on the fine-tuned models, is not uncommon in the misinformation research domain and can be attributed to overfitting.
> - Performance of Zero-Shot Models: In contrast, the zero-shot performance of FLAN-T5 and GPT-3.5-turbo does not come close to 100%, as can be observed in Figures 6 and 7.
> - Impressive Zero-Shot Performance: It's noteworthy that the zero-shot performances of FLAN-T5 and GPT-3.5-turbo are commendable, both in and out of distribution. This contrasts significantly with the performances of our fine-tuned SOTA detectors.
>
>
> (3) Another major issue is that there is no outside evidence to use when assessing misinformation. For example, using the example in Figure 4: how is ChatGPT supposed to know that SF had 2x instead of 5x deaths? It seems without retrieval and outside knowledge this task is ill-formed, and is nearly meaningless - the only way to tell is to try to tell patterns in how the text is written and other heuristics (essentially similar to the watermarking style or chatGPT detection papers). I would recommend checking the line of work on retrieval based fact checking such as FEVER if the authors are not familiar if that's the angle they want, or otherwise focus on those that look at identifying LLM generated text (see references) and use some of them as baselines.
>
> Thank you for your insightful comments. Let us address your concerns:
>
> - Pattern Recognition vs. Fact Retrieval: Our approach focuses on the LLM's ability to detect misinformation through pattern recognition rather than information retrieval. Misinformation, by nature, often manipulates truths in subtle ways, making pattern recognition crucial. The goal is to detect whether a piece of information is 'likely true' based on the patterns recognized from the model's training data, rather than fact-checking it against a database.
> - Limitations of Information Retrieval: While information retrieval can be a potent tool for fact-checking, relying solely on misinformation detection presents challenges. Real-world events like pandemics or natural disasters rapidly produce vast amounts of new information. This dynamic nature means that maintaining a current and comprehensive knowledge base for fact-checking becomes nearly impossible, especially given the many languages and domains involved.
> - Scope of Misinformation Detection: Misinformation detection and fact-checking, although related, are distinct tasks. The former seeks to identify patterns and inconsistencies in the content, while the latter verifies factual accuracy against a known database.
> - Illustrative Example: Consider a statement like "The capital of the United States is Los Angeles." While ChatGPT may not have a real-time database to verify this against, it can recognize that this statement deviates from the dominant pattern in its training data, which associates the U.S. capital with Washington, DC. Thus, it can flag this statement as "likely misinformation."
> - Future Work and Integration: We acknowledge the value of information retrieval and fact-checking methodologies, such as those in the FEVER line of work. In future iterations of our research, we will consider integrating pattern recognition with retrieval-based systems to create a more robust solution. For this current study, however, our focus remains on the LLM's ability to detect misinformation through the patterns it recognizes from its training data.
>
> Thank you for pointing us towards the mentioned references; we appreciate the recommendation and will explore these in-depth for future work.
>
>
> (4) The paper has a significant amount of presentation flaws, such that it will require a major rewrite to be publication ready, with many updates to figures and sections.
>
>
> - Thanks for reading carefully and listing many grammatical and presentation errors in the paper. We will make a thorough revision and go through a professional proof-reading to improve the presentation quality.
>
>
> (5) Minor: FIKE-Human-AI dataset does not seem to be open source and is only “available on upon request”
>
> - Sorry for any misunderstanding. We want to clarify that our codebase and dataset are openly available at: https://anonymous.4open.science/r/FightingFireWithFire. It's not upon request.
>
>
>
> Questions For The Authors:
>
> (1) Lines 344-356: what are the “disagreed generated texts”? Ones where they both don’t pass NLI verification?
>
> - Thank you for pointing that out. By "disagreed generated texts", we refer to instances where there's a mismatch in the outputs of FLAN-T5 and GPT-3.5-turbo. Specifically, one might claim a piece of news is consistent with its original, while the other might say otherwise. To clarify, AI-generated real news is expected to logically align with its original human-written counterpart, while AI-generated fake news should not.
>
>
> (2) If you filter by NLI verification it’s going to be a very easy dataset, so I’m confused,.
>
> - We appreciate your concern. Filtering by NLI verification, especially through the consensus of two distinct LLMs, ensures robustness rather than simplicity. Our aim is to guarantee that synthetic real news remains logically coherent with its original source, whereas the fake news is divergent. Despite maintaining logical coherence, our generated fake news is subtly crafted, posing challenges even for advanced detectors (See Appendix,Table 5 and 6). This complexity mirrors real-world misinformation that often intertwines facts with falsehoods, making detection far from trivial.
>
>
> (3) Further, what does “we confirmed that … they logically entail” mean? You manually checked 40k instances or had an NLI system do it?
>
> Thank you for seeking clarity. To elucidate:
>
> - We conducted a validation process for the generated news content. For the AI-generated real news, we ensured that its content logically aligns with or follows from its corresponding human-written article. This means, for any assertion made in the AI-generated real news, it can be reasonably derived from the content of the human-written article it is paired with.
>
> - On the other hand, for the AI-generated fake news, we ensured the opposite. That is, its content does not logically align with or follow from its corresponding human-written article. Essentially, the fake news content would present assertions or claims that couldn't be logically deduced from the original human-written content.
>
> Both these validation steps were done using a combination of manual checks and logical assessments function to ensure accuracy.
>
>
> (4) Did you filter all datasets or just the new FIKE dataset?
>
> - To address the 40k instances: we employed a hybrid approach. The majority of the validation was carried out using LLMs. However, to ensure the highest degree of accuracy, we conducted manual checks on a subset of these instances. This dual-method allowed us to both efficiently handle the vast volume of data and maintain the integrity of our results, safeguarding against potential misjudgments by the automated system. Central to this approach is our overarching goal: to develop a systematic process that aids future research, especially when confronted with the inherent large amounts of data required to build robust and generalizable models in the misinformation domain.
>
>
>
> (5) Did you filter all datasets or just the new FIKE dataset?
>
> - We applied filtering to all datasets utilized in our research, not just the FIKE dataset. This rigorous process ensures the consistency and quality of our data across the board, setting a strong foundation for our findings and analyses.
>
>
> (6) Missing References:
>
> - Thank you for the suggestion. These papers are relatively new and are not required per EMNLP policies to be cited. However, we will include in our reference.
>
>
> Typos Grammar Style And Presentation Improvements:
>
> Table 2 has many formatting errors, the vertical space is too compressed. Also, lots of negative vspace between captions, the text and figures are running into each other. [Accepted/thank you, we will correct this.]
> Similarly with many of the \paragraph tags like “Prefix Prompt Text Generation” on line 89. [Accepted/thank you, we will correct this.]
> Many parts of Figure 3 I cannot understand: what are MiN, MaJ, CRiT? Etc. Although they are explained later, they should be in a caption or in the figure. [Accepted/thank you, we will correct this.]
> What is dEFEND\C on line 229? The reference for it is several pages later [Accepted/thank you, we will correct this.]
> Table 3 is confusing, could you list the prompts somewhere? Currently they are descriptions of prompts, which is hard to understand and would be simpler to just write the actual prompt. [Acccpted/thank you, we will correct this.]
> Figure 4 the critical example seems like it may be incorrect? It reads the same as the human written post (2x more drug overdoses is the same as more drug overdose deaths, and both complain about the lack of focus on drug overdoses) so it doesn’t seem like a critical difference. [Here are Clarifications]
>
> Fake news often skillfully weaves a false narrative around a kernel of truth, as illustrated in Figure 4. To elucidate, here are the distinctions between Text [Minor] and Text [Critical] itemized:
>
>
> (1) Precision:
> - Text [Minor]  specifies a ratio: For every COVID death, there were five drug overdose deaths.
> Text [Critical] only states that drug overdose deaths exceeded COVID deaths without providing a specific ratio or magnitude.
>
> (2) Tone:
> - Text [Minor] is more factual in its description.
> - Text [Critical] uses a more urgent tone with phrases like "complete disaster" and "epidemic."
>
> (3) Emphasis:
> - Text [Critical] uses exclamation marks ("!") which suggests a heightened sense of urgency compared to Text [Minor].
>
> (4) Call to Action:
> - Text [Critical] explicitly calls for immediate action with phrases like "We need to take action NOW."
>
> (5) Critique of Politicians:
> - Text [Critical] is more direct in its critique, accusing politicians of "turning a blind eye."
>
>
> Figure 5 should not be a line plot, since a change in the x-axis does not have any semantic meaning (is a categorical variable rather than continuous). I would use a bar plot or table. It’s also very difficult to understand the differences with so many lines and little space between them. [Accepted/thank you, we will correct this.]
> What does lines 453-455 mean? ChatGPT is worse than customized detectors but also better?
>
> - Thank you for your query regarding lines 453-455. Let me clarify the distinction and the results presented:
> Customized Detectors: These are detectors models, such as DEFEND/C specifically designed to identify misinformation that we didn’t fine-tune in F3. ChatGPT outperforms these by 15%.
> Fine-tuned Detectors: These are existing pretrained transformer models, such as BERT  that we further trained or "fine-tuned" on F3 DEV misinformation dataset to enhance their detection capabilities. The results show that while GPT-3.5-turbo is good at self-detection, it's still 17% less effective than these fine-tuned detectors.
> In essence, while GPT-3.5-turbo is better than both FLAN-T5 and customized detectors, it does not surpass the performance of detectors that have been specifically fine-tuned for the task. I hope this clears up the distinction and the results presented
>
>
> Line 456 “matching” is not usually the word to be used when something has 2x the score. [Acccpted/thank you, we will correct this.]
> Generally, sections 7.1 and 7.2 would be benefitted by more takeaways and less specific number details. I was a little overwhelmed with numbers and wasn’t sure what the main takeaway was from those sections.  [Acccpted/thank you, we will consider this recommendation.]
>
>
> - We would like to thank the Reviewer for reading our paper so thoroughly and detailing recommendations for improvement. Our intention with the figures is to encourage understanding and not create confusion. Since it did the opposite, we will create better figures and incorporate your recommendations. All recommendations will be reflected in our final draft. Thank you.

---

### Official Review · Reviewer_r37Z · 2023-08-04

**Soundness:** 3

**Excitement:**

4: Strong: This paper deepens the understanding of some phenomenon or lowers the barriers to an existing research direction.

**Paper Topic And Main Contributions:**

According the to abstract, in this paper the authors propose a strategy to identify misinformation with large language models (LLMs). However, I believe that the real value of this paper lies in the experiments and collected data. The strategy itself would rely on the assumption that the model should have access to the correct information.

The authors conduct a very interesting and balanced study in which they test how LLMs (1) generate misinformation (with different strategies), to create test data, and (2) how LLMs can detect misinformation in (a) texts written by humans vs generated by LLMs, and (b) texts written before the cutoff point for GPT-3.5-turbo (Sept. 2021) and after the cutoff point.

The authors test two language models: GPT-3.5-turbo and FLAN-T5, analyze their performance in different setups (including different domains), and release the collected data which may be utilize to further research. The experiments are well described and executed while the RQs are compelling.

**Questions For The Authors:**

Question A: Have you inspected the minor/major/critical perturbations?

Question B: Did you analyze the cases where the systems failed? Did you see any patterns, that is were the systems failing on some specific texts? (beyond what is already discussed as a part of the design).

Question C: How would this work if the information being assessed is not in the knowledge graph? How is the model supposed to evaluate it?

**Reasons To Accept:**

- The paper presents an array of interesting experiments: pre-training vs post-training data; human-written vs AI-generated texts, different severity levels (minor, major, critical), different domains, two LLMs being tested; the researchers also explore the quality of AI-generated text in a sufficient way.

- The RQ are interesting. I found the issue of misinformation identification in AI-generated vs human-written text especially timely and compelling.

- The paper is well written, includes necessary references, most necessary information is already reported.

**Reasons To Reject:**

- It is unclear how this would work if the information is not encoded in the model (similarly to a naive human rater). The authors make an assumption here which is likely not true (why would a model identify something as correct or incorrect without access to the correct answer),

- While the authors divide the data into pre- and post-training, making an assumption that whatever is post-training was not processed by the model, it seems to be also important to acknowledge the potential effect of RLHF.

- If I were to add something, it would small scale manual evaluation of the quality of generated texts, and error analysis. This could share some light on why some assessments are wrong.

**Reproducibility:**

4: Could mostly reproduce the results, but there may be some variation because of sample variance or minor variations in their interpretation of the protocol or method.

**Reviewer Confidence:**

3: Pretty sure, but there's a chance I missed something. Although I have a good feel for this area in general, I did not carefully check the paper's details, e.g., the math, experimental design, or novelty.

**Typos Grammar Style And Presentation Improvements:**

line 48:  "but failed to provide complete understanding" --  understanding of what? Part of text seems to be missing.
line 7: "emergent abilities of LLMs" -- What kind of emergent abilities? I would advise the authors to reword this into something like "leveraging the capabilities of LLMs".

The actual data when GPT-3.5-turbo was used should be added.

---

> ### Author Rebuttal · Authors · 2023-08-29
>
> Reasons To Reject:
>
> (1) It is unclear how this would work if the information is not encoded in the model (similarly to a naive human rater). The authors make an assumption here which is likely, not true (why would a model identify something as correct or incorrect without access to the correct answer).
>
> - We respectfully point out the critical misunderstanding here that a model can answer true/false only when it has access to ground truth. This is incorrect. This misunderstanding seems to yield all the following confusions. LLMs like GPT-3.5-turbo are trained to recognize patterns in the information they've been trained on. The model's response is not about accessing a factual database or knowledge graphs but about recognizing patterns in language and information that align with what's "likely true" based on its training. For example: Let's consider the statement "The capital of the United States is XYZ." Even if the model has never explicitly been fed the information that "The capital of the United States is Washington, DC," it may have seen numerous examples and patterns throughout its training data that align with this fact. If a user claims, "The capital of the United States is Los Angeles," the model, therefore, using its pattern recognition, can predict this statement as "likely false" because it contradicts the overwhelmingly prevalent pattern in its training data that associates the U.S. capital with Washington, DC. This is possible even if LLM does not have access to facts databases or knowledge graphs. In essence, while LLMs do not "know" facts, they have a well-honed ability to recognize and predict patterns, making them valuable tools for identifying inconsistencies or potential misinformation in the context of real vs. fake news.
>
> (2) While the authors divide the data into pre- and post-training, making an assumption that whatever is post-training was not processed by the model, it seems to be also important to acknowledge the potential effect of RLHF.
>
>
> - Thank you for highlighting the importance of RLHF. We would like to clarify a few points:
> Retention Capability: When accessed through the API, GPT-3.5-turbo, like other transformer-based LLMs, does not have the ability to retain new information. This ensures that the model does not "remember" or directly process data that it encounters post-training.
> Impact of RLHF: We concur with your observation. If GPT-3.5-turbo was fine-tuned or influenced by RLHF after its initial training, it could indeed affect the model's behavior or outputs, regardless of whether it has directly seen the post-training data.
> Training Cutoff and RLHF: However, we would like to emphasize that the version of GPT-3.5-turbo used in the F3 study was trained and subsequently fine-tuned with RLHF BEFORE the September 2021 cutoff. This is in line with OpenAI's documentation, which provides a clear distinction on the model's training timeframe. You can find more details about the cutoff date at https://platform.openai.com/docs/models/gpt-3-5.
> Therefore, our decision to segment the data into pre- and post-training, given the above clarifications, aims to ensure a rigorous and robust evaluation process. We appreciate your attention to detail and hope this addresses your concerns.
>
>
>
>
>
>
> (3) If I were to add something, it would small scale manual evaluation of the quality of generated texts, and error analysis. This could share some light on why some assessments are wrong.
>
> - Thank you for your thoughtful suggestion. We appreciate the recommendation to include a small-scale manual evaluation of the quality of generated texts and an error analysis. We would like to point out that we have already conducted a small-scale manual evaluation to assess the contextual similarity between the original human-written texts and the texts generated by our Language Learning Model (LLM).
>
> - In our small evaluation, we sampled 30 AI-generated texts and employed crowdsourcing for assessments. We presented the evaluators with the original human-written text alongside our LLM-generated text. The evaluators were then asked to rate the extent of contextual similarity between the two texts on a scale from 1 to 10.
>
> - In this human evaluation, the median score for contextual similarity was 7.5, and the first quartile score was 6.3. Based on these results, we can confidently assert that the quality of our data is fair, in terms of maintaining contextual similarity.
>
> - Although this evaluation was not included in the initial submission due to its small-scale nature, we believe it also addresses your concerns about the sufficiency and credibility of our data quality. We are open to incorporating this result in appendix of the revised paper for further clarity.
>
> Questions For The Authors:
>
> (1) Question A: Have you inspected the minor/major/critical perturbations?
>
> Yes, we thoroughly examined the minor, major, and critical perturbations through rigorous testing and accuracy inspection.
>
> - Rigorous Testing: Before finalizing our methodology, we conducted rigorous testing on various prompts to ensure optimal results in terms of information generation.
> - Accuracy Inspection Based on Perturbation Levels: For example, for fake news, we did the following
> - Minor Perturbations: Our minor fake news prompt was crafted and inspected for introducing subtle changes to real news, which may not be easily identifiable as misinformation.
> - Major Perturbations: The major perturbation prompt was designed and inspected for eliciting more significant and clear alterations in real news, but still falling short of being extreme.
> - Critical Perturbations: With the critical perturbation prompt, we created and inspected for major, easily recognizable, and impactful deviations from the original real news.
>
> This systematic approach to perturbation levels allowed us to effectively gauge the extent and nature of alterations in the generated text, ensuring robustness in our analysis.
>
>
>
> (2) Question B.1: Did you analyze the cases where the systems failed?
>
> Absolutely, we did analyze instances where the system failed:
> - GPT-3.5-turbo's Self-Explanation: We prompted GPT-3.5-turbo to elaborate on its decisions (whether it identified text as "entail" or "not-entail").
> - Manual Inspection: Whenever GPT-3.5-turbo's explanation for NLI was classified as "other", we undertook a manual inspection. An illustrative example has been provided below that contrasts a human-written text with a critical perturbation-based AI-generated version.
> However, due to the constraints imposed by the conference page limit, we needed help to delve deeper into these failed cases. For future endeavors, a meticulous analysis of these failures can be pursued to enhance generation and detection capabilities. We hope this clarifies.
> - Human-written [Real]:
> “a new study finds that people with depression are less likely to use activities to help regulate their moods. this is something that is even more difficult to do during the covid-19 pandemic. all data and statistics are based on publicly available data at the time of publication. some information may be out of date. visit our coronavirus hub and follow our live updates page for the most recent information on the covid-19 outbreak. the study which now appears in jama examined a selection of a.”
> - AI-Generated [Fake]:
> “A new study has revealed that people with depression are less likely to use activities to regulate their moods, but that's not all. The study, which now appears in JAMA, also found that people with depression are more likely to be infected with COVID-19. The shocking revelation has raised many questions about the relationship between depression and COVID-19.
> The study examined a selection of individuals who had tested positive for the virus and found that those with depression were 70% more likely to contract the virus than those without depression. The study also found that people with depression were less likely to follow guidelines such as wearing masks and social distancing.
> The findings of this study have led to concerns about the mental health of those affected by the virus. With many people struggling to cope with the stress and anxiety caused by the pandemic, the added risk of contracting the virus has only added to their burden.
> Experts are now calling for increased support for those with depression during the pandemic, including access to mental health services and resources to help them cope with the challenges they are facing.
> As the pandemic continues to affect people around the world, it is essential that we take steps to address the mental health challenges that are arising. By providing support and resources to those who need it most, we can help ensure everyone can stay healthy and safe during these difficult times.”
> NLI Label: Other
> - GPT-3.5-Turbo Explanation:
> Premise: A new study finds that people with depression are less likely to use activities to help regulate their moods. This is something that is even more difficult to do during the COVID-19 pandemic.
> Conclusion: There is no clear conclusion provided in the given text. However, it can be inferred that people with depression may face additional challenges during the COVID-19 pandemic. Therefore, the relationship between depression and COVID-19 may need to be further explored to understand better the potential risks associated with COVID-19.
>
>
>
>
>
>
> (3) Question B.2: Did you see any patterns, that is, were the systems failing on some specific texts? (beyond what is already discussed as a part of the design).
>
> - Indeed, we observed patterns in the system's performance. While the majority of the behaviors fall within our earlier design discussion, one distinct pattern emerged: GPT-3.5-turbo notably found it challenging to classify fake AI-generated news that had elements of truth interwoven. These intricately designed pieces aim to deceive non-critical human readers by blending falsehood with genuine facts. This observation underscores the importance of refining AI techniques to handle such nuanced misinformation scenarios better.
>
>
>
>
> (4) Question C.1: How would this work if the information being assessed is not in the knowledge graph?
>
>
> - Our work utilizes pre-trained LLMs that discern patterns from real and fake news, which enables them to generalize to unseen data. The models employed in our experiments neither rely on nor use knowledge graphs.  While it is also possible to build Knowledge-graph enhanced LLMs, all experiments in our setting used LLMs that do not require nor use knowledge graphs. The ability to perform as an expert such as a knowledge graph would be one of the many emergent abilities of new-age LLMs which we aim to evaluate and investigate in F3.
>
>
>
>
> (5) Question C.2:  How is the model supposed to evaluate it?
>
> - The LLM evaluates information based on its training data, making inferences anchored to its prior knowledge.
>
>
> Typos Grammar Style And Presentation Improvements:
>
> (1) line 48: "but failed to provide complete understanding" -- understanding of what? Part of text seems to be missing. line 7: "emergent abilities of LLMs" -- What kind of emergent abilities? I would advise the authors to reword this into something like "leveraging the capabilities of LLMs".
>
>
> - Thank you for your suggestion
> A recent work (Bang et al., 2023) showed the potential of ChatGPT for detecting misinformation in limited data settings but failed to provide an extensive analysis using larger datasets or comparing its efficacy against state-of-the-art (SOTA) detectors.
>
> - In response to this challenge, we propose the “Fighting Fire with Fire” (F3) strategy, leveraging emergent capabilities of LLMs such as in-context learning and step-by-step reasoning to combat human-written and AI-generated misinformation through prompt engineering.

---

### Official Review · Reviewer_h75L · 2023-08-05

**Soundness:** 3

**Excitement:**

3: Ambivalent: It has merits (e.g., it reports state-of-the-art results, the idea is nice), but there are key weaknesses (e.g., it describes incremental work), and it can significantly benefit from another round of revision. However, I won't object to accepting it if my co-reviewers champion it.

**Paper Topic And Main Contributions:**

This paper studies misinformation generation detection and detection by prompting large language models. The authors propose F3 prompting technique for synthetic real news and fake news generation based on perturbation and paraphrasing with ChatGPT and filter data with NLI techniques. The authors also experiment with different misinformation detection methods and find that ChatGPT demonstrates superior effectiveness in detecting misinformation generated by itself.

**Questions For The Authors:**

- L335. The NLI task for verifying the quality of the generated information is to test whether the syntactic news entail the human-written news. Why not test it in the reversed order?

**Reasons To Accept:**

- The prompting technique for misinformation generation and detection is novel. Comprehensive experiments are helpful for understanding the proposed method.

- The finding that ChatGPT can better identify misinformation generated by itself is insightful.

- The dataset constructed using ChatGPT could facilitate future research on misinformation detection.


**Reasons To Reject:**

- The position of this work in related work is not well discussed. For example, how is the technique used in this work different from previous work? What are the uniqueness of the proposed dataset compared to existing resources?

- Their lacks a rigorous evaluation on the quality of the misinformation dataset constructed with ChatGPT. It’s not clear how text similarity and NLI can ensure the data quality.

- The writing can be improved for better clarity. See “Typos Grammar Style And Presentation Improvements”.


**Reproducibility:**

N/A: Doesn't apply, since the paper does not include empirical results.

**Reviewer Confidence:**

4: Quite sure. I tried to check the important points carefully. It's unlikely, though conceivable, that I missed something that should affect my ratings.

**Typos Grammar Style And Presentation Improvements:**

- L363, the section number is missing.

- Although the description about the proposed prompts is introduced, the specific prompt text seems to not have been given in the paper.

- L235: it’s inappropriate to name the instructive text for misinformation generation as a “verbalizer”. Verbalizers refer to mappings between LM outputs and task labels.

- The figures/tables are not on the same page with their context, making it hard to read.

- There are too many numbers mentioned in the experimental results. The authors can improve its clarity by only presenting the most important ones and making better correspondence with the numbers in figures/tables.

---

> ### Author Rebuttal · Authors · 2023-08-29
>
> Reasons To Reject:
>
> Q1. The position of this work in related work is not well discussed. For example, how is the technique used in this work different from previous work?
>
>
> F3's uniqueness:
>
> GENERATION:
>
> 1. Novel Prompting Techniques for Generation:
> - We use both paraphrased-based and perturbation-based prompts to generate both real and fake news, respectively with GPT-3.5. Using these novel prompting techniques, we show that GPT-3.5 guardrails could be bypassed by malicious actors.
>
> 2. Hallucinated text detection:
> - As LLMs have been shown to hallucinate (ie, generate texts that are unfaithful from the prompt), we made sure that the real news generated was in fact real news and the same for fake news. We employed text entailment strategies for the task of detecting and removing hallucinated texts. In the spirit of fighting fire with fire, we use GPT-3.5 to detect hallucinated texts. The hypothesis here is:
> LLM-generated real news must entail Human-written real news. In order words, they should be semantically consistent
> LLM-generated fake news must NOT entail Human-written real news. In order words, they should not be semantically consistent, else, GPT-3.5 has hallucinated and generated another real news, instead of fake news.
>
> DETECTION
>
> 4. Novel Prompting Techniques for Detection:
> - Again, in the spirit of fighting fire with fire, we employ state-of-the-art reasoning methods such as Chain of Thought (CoT), Auto Chain of Thought with confidence reasoning, and other variants of such prompts to detect real vs. fake news by both human and LLM authors.
>
> 5. LLM vs. SOTA Fake News Detectors:
> - We compare LLMs, GPT-3.5 and Flan-T5 to SOTA fake news detectors to leverage the abilities of such LLMs for accurate fake news detection.
>
>
>
>
>
>
> Q.2. What is the uniqueness of the proposed dataset compared to existing resources?
>
> It is the first misinformation dataset that evaluated and removed AI-generated content subjected to  'hallucination'—where Large Language Models (LLMs) produce text unfaithful to the prompt. We ensure that real is actually real and fake news is actually fake [2]. While rarely prior studies [1,2] invesitigated AI-generated misinformation, they did not rigorously verify the fidelity of such generated content [3,4,5]. Please see a comparison of our dataset an other dataset below:
>
> (1) Unique Generation Techniques in FIKE:
> - It uses paraphrase-based prompt engineering for synthetic real news generation.
> - Employs perturbation techniques for synthetic fake news generation
> - Features three AI-generated news variations: Min, Maj, Crit.
>
> (2) Cover real-world misinformation setting:
> - Incorporates both human and AI-generated misinformation content.
> - Includes diverse sources: articles and social media.
>
> (3) Covers more than 1 misinformation topic domain:
> - Spans multiple areas, such as COVID-19 and politics.
>
> (4) Up-to-date & OOD Studies:
> - Covers misinformation from 2021-2023.
> - Allows for in and out-of-distribution studies.
> - Other datasets, especially concerning politics and COVID-19, were created pre-2021  [3,4,5].
>
> (5) Unprecedented Public Availability:
> - The first public dataset with misinformation by recent LLMs, such as GPT-3.5-turbo.
> - Publicly available, unlike many misinformation datasets that are available by request
>
> [1] Zhou, J., Zhang, Y., Luo, Q., Parker, A. G., & De Choudhury, M. (2023, April). Synthetic lies: Understanding ai-generated misinformation and evaluating algorithmic and human solutions. In Proceedings of the 2023 CHI Conference on Human Factors in Computing Systems (pp. 1-20).
>
> [2] Ji, Z., Lee, N., Frieske, R., Yu, T., Su, D., Xu, Y., ... & Fung, P. (2023). Survey of hallucination in natural language generation. ACM Computing Surveys, 55(12), 1-38.
>
> [3] Murayama, T. (2021). Dataset of fake news detection and fact verification: a survey. arXiv preprint arXiv:2111.03299.
>
> [4] Su, Q., Wan, M., Liu, X., & Huang, C. R. (2020). Motivations, methods and metrics of misinformation detection: an NLP perspective. Natural Language Processing Research, 1(1-2), 1-13.
>
> [5] Oshikawa, R., Qian, J., & Wang, W. Y. (2018). A survey on natural language processing for fake news detection. arXiv preprint arXiv:1811.00770.
>
>
>
>
>
>
>
>
>
> Q.3 Their lacks a rigorous evaluation on the quality of the misinformation dataset constructed with ChatGPT. It’s not clear how text similarity and NLI can ensure data quality.
>
>
> - Given a piece of human-written text (real news), T, we prompt an LLM, to generate real news, T’_real  and fake news, T’_fake, such that T’_real is similar to T and T’_fake is dissimilar to T. Due to LLM’s ability to sometimes generate texts, unfaithful to the prompt, we define an entailment model - N(.), such that for LLM-generated real news, T’_real should entail T, and for fake news, T’_fake should Not-entail T. Therefore, using the entailment model, N(.), we can access logical consistency between the original (human-written) and the generated texts, thus removing hallucinated LLM-generated texts.
> - To further understand the data, ensure that the N(.) model detected and removed hallucinated texts, we measured the contextual and semantic consistency of the LLM-generated texts. We use BERT and BLEURT-Scores to measure the semantic contextual and semantic consistency, respectively. The hypothesis here is that real news should demonstrate more semantic and contextual alignment than fake news (See Figure 14 in Appendix). This is because misaligned contextual consistency may not impact fake news because fake news should have a looser constraint than real news.
> - BERT evaluates AI-generated text by analyzing its contextual word understanding. It measures how closely the generated text aligns with the original, especially distinguishing contextual consistency of AI-generated real from fake news. BLEURT-20, used mainly for translation comparisons, checks if the AI content mirrors the original's structure. Both BERT and BLEURT-20 ensure that AI-generated news stays true to the original's context and meaning. According to the metrics, achieving an average score of over 90% is considered high-quality generated texts in terms of semantics and context. Thus, using these metrics, after removing the generated texts that failed the entailment assessment, we achieved an average score of 94% for BERTscore, with difference in scores between real vs. fake for the BLEURT-score, indicating intuitive semantic changes between real and fake news.
>
> - In addition to the computational metrics and entailment models mentioned, we also conducted a small-scale human evaluation to validate the quality of our misinformation dataset generated with ChatGPT. In this evaluation, we sampled 30 AI-generated texts and used crowdsourcing to assess their contextual similarity to the original human-written texts. Participants were presented with both versions and were asked to rate the contextual similarity on a scale from 1 to 10. The median score received was 7.5, and the first quartile was 6.3, indicating a fair level of quality in maintaining contextual similarity. Importantly, this human assessment aligns well with our computational metrics. Although this evaluation was not included in the initial submission due to its small-scale nature, we believe it also addresses your concerns about the sufficiency and credibility of our data quality. We are open to incorporating this result in appendix of the revised paper for further clarity.
>
>
>
>
> Q.4 The writing can be improved for better clarity. See “Typos Grammar Style And Presentation Improvements”.
>
> - We thank the reviewer for letting us know of the typos. These recommended changes will be reflected in our final draft.
>
>
>
> Questions For The Authors:
>
> Q1. L335. The NLI task for verifying the quality of the generated information is to test whether the syntactic news entail human-written news. Why not test it in the reversed order?
>
> - We thank the reviewer for this insightful question. While it is valid to test the reversed order of the entailment analysis, the answer will still remain the same in this case. This is because for GPT-3.5 and Flan-T5 models, order does not matter. Which is synonymous to the communicative law of addition, where A + B = B + A. However, through your insightful question, we realize that we can make our test even stricter, such that it is more synonymous to the communicative law of division, where A ÷ B ≠ B ÷ A. In this instance order does matter. This means that we can use the set theory, where we check if A, LLM real news is a subset of B, human-written real news and C, LLM fake news is NOT a subset of B. Thus, we will use such techniques to detect hallucinated text more strictly in future work. Thank you again.
>
>
>
>
> Typos Grammar Style And Presentation Improvements:
>
> L363, the section number is missing.
> Although the description about the proposed prompts is introduced, the specific prompt text seems to not have been given in the paper.
> L235: it’s inappropriate to name the instructive text for misinformation generation as a “verbalizer”. Verbalizers refer to mappings between LM outputs and task labels.
> The figures/tables are not on the same page with their context, making it hard to read.
> There are too many numbers mentioned in the experimental results. The authors can improve its clarity by only presenting the most important ones and making better correspondence with the numbers in figures/tables.
>
> - Thank you for the detailed list of recommended changes, This is very helpful. We understand the confusion behind the use of verbalizer and we will fix this in our final draft. Additionally, we intend to take the very useful recommendation of summarizing most important findings into tables.

---

### Meta-Review · Area_Chair_mHvj · 2023-09-19

**Recommendation:** 4

**Metareview:**

This paper introduces a novel method for misinformation detection using large language models (LLMs), substantiated by extensive experiments. The authors employ a prompting technique and find ChatGPT effective in identifying its generated misinformation. The paper's clarity and dataset quality evaluation have been criticized, but the authors present a reasonably good rebuttal. Despite some remaining concerns around the paper's assumptions, it offers timely and crucial insights into misinformation detection. However, it needs further refinement, particularly in experimental design and results presentation. Despite minor points needing attention, the paper usefully contributes to the misinformation detection field.

---

### Decision · Program_Chairs · 2023-10-07

**Decision:**

Accept-Main

**Comment:**

This paper introduces a novel method for misinformation detection using large language models (LLMs), substantiated by extensive experiments. The authors employ a prompting technique and find ChatGPT effective in identifying its generated misinformation. The paper's clarity and dataset quality evaluation have been criticized, but the authors present a reasonably good rebuttal. Despite some remaining concerns around the paper's assumptions, it offers timely and crucial insights into misinformation detection. However, it needs further refinement, particularly in experimental design and results presentation. Despite minor points needing attention, the paper usefully contributes to the misinformation detection field.